# Personalized Federated Learning with Spurious Features: An Adversarial Approach

**Xiaoyang Wang**                                                                                  *xw28@illinois.edu*
*Department of Computer Science*
*University of Illinois at Urbana-Champaign*

**Han Zhao**                                                                                  *hanzhao@illinois.edu*
*Department of Computer Science*
*University of Illinois at Urbana-Champaign*

**Klara Nahrstedt**                                                                                  *klara@illinois.edu*
*Department of Computer Science*
*University of Illinois at Urbana-Champaign*

**Sanmi Koyejo**                                                                                  *sanmi@cs.stanford.edu*
*Department of Computer Science*
*Stanford University*

**Reviewed on OpenReview:** *https://openreview.net/forum?id=N2wx9UVHkH*

## Abstract

One of the common approaches for personalizing federated learning is fine-tuning the global model for each local client. While this addresses some issues of statistical heterogeneity, we find that such personalization methods are vulnerable to spurious features at local agents, leading to reduced generalization performance. This work considers a setup where spurious features correlate with the label in each client's training environment, and the mixture of multiple training environments (i.e., the global environment) diminishes the spurious correlations. In other words, while the global federated learning model trained over the global environment suffers less from spurious features, the local fine-tuning step may lead to personalized models vulnerable to spurious correlations. In light of this practical and pressing challenge, we propose a novel strategy to mitigate the effect of spurious features during personalization by maintaining the adversarial transferability between the global and personalized models. Empirical results on object and action recognition tasks show that our proposed approach bounds personalized models from further exploiting spurious features while preserving the benefit of enhanced accuracy from fine-tuning.

## 1 Introduction

Federated learning (FL) is a leading framework for clients to collaboratively train a shared global machine learning (ML) model without releasing their local private datasets (McMahan et al., 2017; Kairouz et al., 2019). The jointly trained global model can be further fine-tuned on each client's local dataset to produce personalized (local) models (Fallah et al., 2020; T. Dinh et al., 2020; Li et al., 2021). While existing theoretical and empirical results highlight how personalized models improve accuracy on local data, few works consider what features the personalized models learn from the local dataset. Our motivating hypothesis is that, *not all local features are beneficial.*

*Spurious features*, which correlate with labels in some environments but do not generalize across all environments, are ubiquitous (Geirhos et al., 2020; Singla & Feizi, 2022). For example, for activity recognition, a rocky cliff can correlate with climbing activity (Nam et al., 2020); for classification, flowers can be a spurious

Table 1: Requirement comparison. Our approach limits personalization from increasing the entanglement level to spurious features without requiring access to multiple environments or their indexes. Environment indexes are part of the label of data samples, indicating which environment the data sample comes from.

| Method | Environments | Environment Indexes |
|---|---|---|
| Invariant risk minimization (Arjovsky et al., 2019) | Must-have | Must-have |
| Distributionally robust optimization (Sagawa et al., 2020) | Must-have | Must-have |
| Just-train-twice (JTT) (Liu et al., 2021) | Must-have | Need-not |
| Ours | Need-not | Need-not |

feature in butterfly images (Singla & Feizi, 2022); for recognition, hair color may confound the gender of face images (Liu et al., 2021). These examples are illustrated with images in Appendix A. In these examples, the predictive performance of a model that entangles spurious features can degrade in environments where the spurious correlation no longer holds (e.g., for the activity detection example, consider climbing on icy walls). Such non-uniform performance across environments due to the entanglement of spurious features has led to issues in robustness and fairness, among others (He et al., 2019; Nam et al., 2020; Liu et al., 2021).

This paper considers a setup where each user holds training samples from their own environment but may move to other environments with a deployed model. Such a setup applies to federated learning systems with diverse and heterogeneous clients, who tend to collect training samples around their current environment and may later travel with their mobile devices. Examples include mobile augmented reality applications (Cao et al., 2022), where client-side distributions change spatially. In such cases, although spurious features correlate with the label on each client's *training environment*, aggregating the clients over different environments (i.e., the *global environment*) contributes to mitigating the spurious correlations.

In FL, even if the spurious correlations in the global environment can be diminished, naive local fine-tuning surely increases the entanglement level of personalized models to spurious features of their local training environments. Thus, learning a personalized model for each client without risking entanglement to spurious features is non-trivial. Common methods that aim to disentangle spurious features from models (Wang et al., 2019; Sagawa et al., 2020; Liu et al., 2021; Wang et al., 2022b) require full access to multiple environments and are, therefore, not applicable to our federated setting. On the other hand, although using the off-the-shelf global model in local agents without fine-tuning avoids further spurious correlations, the performance on each local client's dataset can be sub-optimal.

To this end, we propose a novel method to limit personalization from increasing the entanglement level to spurious features without requiring access to multiple environments (Table 1) and subsequently improve the generalization performance of personalized models measured by the *accuracy disparity* (Zhao & Gordon, 2022)–the difference between a model's accuracies across environments. We call the difference between the entanglement level of the global and personalized models *entanglement deviation*. Then, our method uses the *adversarial transferability* between the global and personalized models as a proxy to bound the entanglement deviation. The adversarial transferability is measured by the percentage of adversarial examples generated by the global model that also "fool" personalized models. The intuition is that if two models entangle features in the same way, the adversarial examples are more likely to transfer from one to another and vice versa. Based on this intuition, we propose the following hypothesis:

> *If personalized models increase their entanglement to spurious features, fewer adversarial examples generated by the global model transfer to personalized models.*

Note that our approach does not require entangling spurious features to be an exclusive cause of decreasing the adversarial transferability and remains effective as long as such a connection exists. Empirically, we observe that the adversarial transferability between the global and personalized models often decreases when personalized models entangle spurious features. In such cases, the entanglement deviation will increase (Section 3), validating our hypothesis. However, naively including adversarial examples with flipped labels (i.e., mispredictions) to fine-tune and maintain the adversarial transferability is sub-optimal in bounding the entanglement deviation (Section 3.3). Therefore, we theoretically analyze the connection between the

adversarial transferability and the entanglement deviation of the global and personalized models. In addition, we show that the disparity can be bounded by the entanglement and present conditions under which the disparity upper bound of personalized models can be close to that of the global model (Section 4). Based on the theoretical results, we develop an improved method to bound the entanglement deviation and the accuracy disparity of personalized models (Section 5), thus improving the robustness of personalized models to local spurious features. Our main contributions are:

- We empirically evaluate the vulnerability of personalized models to spurious features in a federated learning setting, highlighting a critical risk of existing personalization methods.

- We theoretically connect the adversarial transferability and the entanglement levels of the global and personalized models to spurious features.

- We develop a method to bound the increased entanglement of personalized models to spurious features by maintaining the adversarial transferability between the global and personalized models.

We conduct extensive experiments to validate the effectiveness of the proposed methods under FL settings. Our experiments on MNIST (Deng, 2012), Coil20 (Nene et al., 1996), CelebA (Liu et al., 2015; Caldas et al., 2018), and biased action recognition (BAR) (Nam et al., 2020) datasets show that the proposed approach reduces the accuracy disparity of personalized models from 18.38% to 3.42%. Our method also preserves the benefit of the enhanced average accuracy from fine-tuning, resulting in 4.48% accuracy improvement in the global environment.

## 2 Related Work

**Personalized Federated Learning.** Fine-tuning is typical for personalizing FL. The meta-learning-based method first trains a global model and fine-tunes the global model locally (Fallah et al., 2020). Other methods using multi-task learning (Li et al., 2021) or Moreau envelopes (T. Dinh et al., 2020) have an interpretation as fine-tuning the local model along with training the global model. Fine-tuning is also compatible with clustering-based methods (Ghosh et al., 2020). FedPAC (Xu et al., 2023) and FEDORA (Wu et al., 2023) further integrate the idea of clustering and fine-tuning: they enable knowledge transfer between personalized models on clients that are similar to each other. There are also approaches investigating fine-tuning using sub-networks (Shamsian et al., 2021) or using a k-nearest-neighbor (kNN) classifier as an augmentation to a local model (Marfoq et al., 2022). Our work focuses on limiting the entanglement of personalized models during fine-tuning via efficient regularization. Therefore, we will compare our approach with other fine-tuning approaches that focus on the loss function design (Li et al., 2021; Xu et al., 2023; Wu et al., 2023). The sub-network and kNN approaches are not direct competitors to our approach, and their combination with our work can be interesting for future work.

**Debiasing Machine Learning Models.** Debiasing is a way to disentangle spurious features from the model. Chi et al. (2021) aims to mitigate the accuracy disparity in regression problems via learning the appropriate representations. A few prior works (Li & Vasconcelos, 2019; Sagawa et al., 2020; Wang et al., 2022b) utilize group labels, which might require human annotation, to debias ML models. Residual learning-based methods (He et al., 2019; Nam et al., 2020; Liu et al., 2021) train a biased ML model and up-weight the residual, which mainly contains samples from under-represented environments that the biased ML model mis-predicts. Inspired by distributional robust optimization (DRO) (Duchi et al., 2016; 2023), the group DRO approach (Sagawa et al., 2020) minimizes the worst-case training loss over a mixture of pre-defined groups via selectively assigning higher weights to underfitted groups. However, a disadvantage of the residual learning-based and the DRO-based approaches is their requirement for precise group annotations (Table 1). A follow-up work, called just-train-twice (JTT) (Liu et al., 2021), further removed this requirement, which is comparable to our work and is included in the experiments. Note that our approach does not require group annotations either. In addition, we find an issue with the approach of "assigning higher weights to under-represented environments during training" in the federated learning personalization step– when each client has very few samples. The samples from under-represented environments can be too few for the model

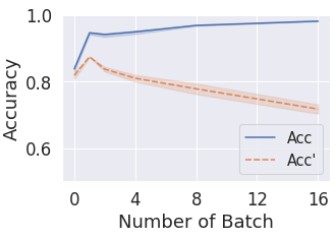
(a) Accuracy disparity.

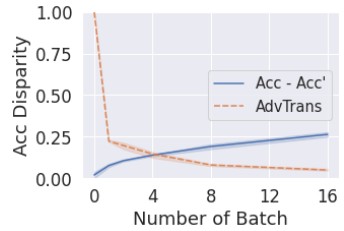
(b) Accuracy disparity correlates with adversarial transferability.

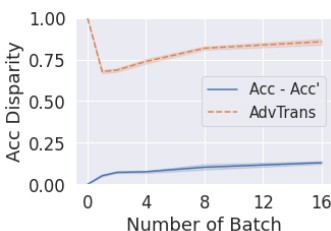
(c) Naively maintaining high adversarial transferability does not help.

Figure 1: An empirical study to evaluate the transferability hypotheses with spurious features. (a) The accuracy of personalized models in the training environments (Acc) and the other environments that are not accessible during training (Acc$'$) with increasing fine-tuning batches. The personalized models entangle the spurious feature and increase accuracy disparities between environments. (b) As the personalized models entangle the spurious feature and increase their accuracy disparity (Acc - Acc$'$), adversarial transferability (AdvTrans) decreases. (c) Naively maintaining high adversarial transferability (AdvTrans) helps mitigate the accuracy disparity (Acc - Acc$'$), but the accuracy disparity still increases from 0.06 to 0.12 along with the increasing adversarial transferability.

to reach a desirable performance even if we assign them higher weights (see experiments in Section 6.2.2). In contrast, our approach does not require access to samples from minority groups during personalization – and we find that our approach does not suffer from such a limitation.

**Adversarial Transferability.** Prior works (Tramèr et al., 2017; Charles et al., 2019) attempt to understand adversarial transferability. A recent work connects adversarial transferability to knowledge transferability in a transfer learning setting (Liang et al., 2021). Our paper builds on this work to show how to solve a new problem–bounding the personalized models' entanglement to spurious features–by maintaining adversarial transferability between the global and personalized models.

## 3   An Empirical Study with Spurious Features

We first perform an empirical study on the entanglement deviation between the global and personalized models in an FL setting to gain some insights into the problem. This section focuses on the MNIST dataset due to the limited space, and Appendix E further provides results on the other datasets. In this study, the personalization method is fine-tuning. Our results highlight the risk of existing fine-tuning-based personalization methods and the difficulty of mitigating the risk. We also highlight the correlation between adversarial transferability and entanglement deviation. We provide additional theoretical analysis in Section 4 for more insights into the observed correlation. The client setup and evaluation protocol in this empirical study are as follows.

**Client Setup.** There are two environments whose combination is the global environment. Each client gets a different set of grayscale MNIST digits, then creates a spurious correlation by coloring the grayscale objects according to their labels differently for each environment (e.g., the red color correlates with digit 0 in the first environment and with digit 1 in the second environment). Then, each client picks a random environment for training, and all clients collaboratively train a global model. After the global model converges, clients further fine-tune the global model in their training environments.

**Evaluation Protocol.** We evaluate the accuracy of personalized models in their training environments (Acc) and the other environments that are not accessible during training (Acc$'$) with an increasing number of fine-tuning steps (i.e., batches). The accuracy disparity indicates entanglement to spurious features, defined as the accuracy difference between the training environment and the environment that is not accessible during training (i.e., Acc - Acc$'$).

### 3.1 Personalization May Exacerbate Entanglement

As can be observed in Figure 1a, the accuracy disparity of personalized models increases within a few fine-tuning batches, deviating from that of the global model. This observation indicates that personalized models gradually increase the entanglement to the spurious feature. Although, in principle, one may resort to early stopping, this is not feasible using a single training environment alone.

### 3.2 Transferability Correlates with Entanglement

Because only the training environment is accessible during personalization, it is infeasible to directly measure personalized models' entanglement or accuracy disparity. To this end, in this section, we focus on methods that implicitly measure and determine to which level personalized models can entangle the spurious feature. Following our hypothesis in Section 1, we consider using the adversarial transferability between the global and personalized models as a proxy for measuring their entanglement deviation.

**Adversarial Attack**   An adversarial attack aims to add a calibrated adversarial perturbation to the input to mislead a model's prediction. A perturbed input is an adversarial example. A common way to calibrate the adversarial perturbation is leveraging the model's first-order gradient w.r.t. the input (Madry et al., 2018; Miyato et al., 2018; Liang et al., 2021). Such a first-order gradient suggests a direction along which the prediction changes significantly. In our setting, the first-order gradient shall direct more perturbations to robust features to mislead the global model, which entangles spurious features less. In contrast, if personalized models entangle the spurious features more, they are more likely to resist the attack that targets the global model by using the spurious features with fewer perturbations. Following this intuition, we employ the projected gradient descent (PGD) attack (Madry et al., 2018) that iteratively calibrates the adversarial perturbation using the first-order gradient. Specifically, at a point $\boldsymbol{x}$ with label $y$, for the global model $f_g$ with loss function $\ell$, at iteration $t+1$, the adversarial example is:

$$\boldsymbol{x}_{\mathrm{adv}}^{t+1} = \mathrm{Proj}_{\|\boldsymbol{x}_{\mathrm{adv}}-\boldsymbol{x}\|\leq\epsilon}(\boldsymbol{x}_{\mathrm{adv}}^{t} + \alpha \cdot \mathrm{sign}(\nabla_{\boldsymbol{x}_{\mathrm{adv}}^{t}}\ell(f_g(\boldsymbol{x}_{\mathrm{adv}}^{t}), y))),$$

where Proj is a projection operator, $\boldsymbol{x}_{\mathrm{adv}}^{0} = \boldsymbol{x}$, $\epsilon$ is the attack budget, and $\alpha$ is the attack step size.

**Adversarial Transferability**   We generate adversarial examples using the global model and collect the ones that cause the global model to mispredict. Then, personalized models make predictions on the collected adversarial examples. The adversarial transferability is defined as the percentage of the collected adversarial examples that also cause personalized models to mispredict. Recalling that the collected adversarial examples embed more perturbation on robust features, personalized models may resist the adversarial examples if they use spurious features to predict.

Figure 1b plots the accuracy disparity and the adversarial transferability during fine-tuning. As personalized models increase their entanglement to spurious features, the accuracy disparity of the personalized model increases and deviates from that of the global model. Then, the adversarial transferability between the global and personalized models decreases. This result empirically validates our hypothesis.

### 3.3 Can Maintaining Transferability Mitigate Entanglement Deviation?

Following the empirical observation of the adversarial transferability and the entanglement deviation, we add the collected adversarial examples with flipped labels (i.e., mispredictions) to the training set during personalization, aiming to maintain the adversarial transferability. However, Figure 1c shows that the accuracy disparity still doubles (0.06 to 0.12), indicating that the entanglement deviation increases, even if the adversarial transferability increases. The following section presents a theoretical analysis, outlining conditions under which maintaining the adversarial transferability helps bound the entanglement deviation.

## 4 Theoretical Insights

In this section, we present a theoretical analysis that supports our hypothesis in Section 1, the experimental results in Section 3.2, and analyze the failure in Section 3.3. Before we proceed to the detailed analysis, some

additional definitions and notations are needed for the presentation (we provide a table summarizing all the notations used in Appendix B to ease the reading).

## 4.1 Setting

**Data Model.** A data sample $\boldsymbol{x} = \text{Cat}(\boldsymbol{x}_r, \boldsymbol{x}_s)$ is a concatenation of robust features $\boldsymbol{x}_r$ (i.e., $\dim(\boldsymbol{x}_r) \geq 2$) and a spurious feature $\boldsymbol{x}_s$. The label is binary, $y \in \{0, 1\}$.

**Environment.** We consider a setting with two environments.[1] On the $i^{\text{th}}$ client, $\mathcal{D}_i$ denotes the joint distribution of $\boldsymbol{x}$ and $y$ in the training environment and $\mathcal{D}'_i$ denotes the future environment of the $i$-th client that is not accessible during training. In addition, we use $\mathcal{D}_{i,r}$ to denote the joint distribution of $\boldsymbol{x}_r$ and $y$ and use $\mathcal{D}_{i,s}$ to denote the joint distribution of $\boldsymbol{x}_s$ and $y$. Our analysis applies to each client's $\mathcal{D}$ and $\mathcal{D}'$, and we shall omit the subscripts $i$ when it is clear from the context.

**Distribution Shift.** Our analysis focuses on conditional distribution shift caused by spurious correlations, which imply $\mathcal{P}_{\mathcal{D}}(\boldsymbol{x}_s \mid y) \neq \mathcal{P}_{\mathcal{D}'}(\boldsymbol{x}_s \mid y)$ but $\mathcal{P}_{\mathcal{D}}(\boldsymbol{x}_r \mid y) = \mathcal{P}_{\mathcal{D}'}(\boldsymbol{x}_r \mid y)$ and $\mathcal{P}_{\mathcal{D}}(y) = \mathcal{P}_{\mathcal{D}'}(y)$.

**Data Generating Process.** We consider a data-generating process that produces a pair of samples with four steps: (1) draw a label $y$ according to $\mathcal{P}_{\mathcal{D}}(y)$, (2) draw a robust feature $\boldsymbol{x}_r$ under $\mathcal{P}_{\mathcal{D}}(\boldsymbol{x}_r \mid y)$, (3) draw a spurious feature according to $\mathcal{P}_{\mathcal{D}}(\boldsymbol{x}_s \mid y)$ and concatenate features $\boldsymbol{x} = \text{Cat}(\boldsymbol{x}_r, \boldsymbol{x}_s)$, and (4) draw a spurious feature with $\mathcal{P}_{\mathcal{D}'}(\boldsymbol{x}_s \mid y)$ and concatenate features $\boldsymbol{x}' = \text{Cat}(\boldsymbol{x}_r, \boldsymbol{x}'_s)$. By the linearity of expectation, the four-step data-generating process does not affect the loss disparity:

$$|\mathbb{E}_{\mathcal{D}}[\ell(f(\boldsymbol{x}), y)] - \mathbb{E}_{\mathcal{D}'}[\ell(f(\boldsymbol{x}), y)]| = |\mathbb{E}_{\mathcal{D}, \mathcal{D}'}[\ell(f(\boldsymbol{x}), y) - \ell(f(\boldsymbol{x}'), y)]|. \tag{1}$$

**Hypothesis Class.** We consider a logistic regression model $f(\boldsymbol{x}) = \sigma(\boldsymbol{w}^\top \boldsymbol{x})$ with a Sigmoid activation function $\sigma$, and a logistic loss $\ell$. Similar to the notations in the data model, we let $\boldsymbol{w} = \text{Cat}(\boldsymbol{w}_r, \boldsymbol{w}_s)$, where $\boldsymbol{w}_r$ includes the weights on robust features $\boldsymbol{x}_r$ and $\boldsymbol{w}_s$ denotes the weight of spurious feature $\boldsymbol{x}_s$. $f_g$ denotes the global model with weight $\boldsymbol{w}_g$ and $f_l$ denotes the personalized model with weight $\boldsymbol{w}_l$. $\boldsymbol{w}_{g,s}$ denotes the global model's weight of the spurious feature.

## 4.2 Preliminaries

**Entanglement.** For a given model $f$ with weight $\boldsymbol{w}$, its entanglement to the spurious feature $\|\boldsymbol{x}_s\|$ is defined as $\boldsymbol{w}_s$. The entanglement $\|\boldsymbol{w}_s\|$ can be further decomposed into a norm term and an angle term: $\|\boldsymbol{w}_s\| = \|\boldsymbol{w}\| \cos\theta$, where $\theta$ is the angle between $\boldsymbol{w}$ and $\text{Cat}(\mathbf{0}, \boldsymbol{w}_s)$. Such a definition of $\theta$ implies that $\theta \in [0, \frac{\pi}{2}]$ and $\cos\theta \in [0, 1]$. Compared to existing definitions such as $\boldsymbol{w}_s = 0$ from the literature on disentangling spurious features (Rosenfeld et al., 2021; Wang et al., 2022a), our definition allows a quantitative measurement of the entanglement/disentanglement to spurious features and includes the standard objective as a special case (i.e., $\|\boldsymbol{w}\| = 0$ or $\theta = 0$).

**Angles** Let $\theta' = \arccos \frac{\boldsymbol{w}_g \cdot \boldsymbol{w}_l}{\|\boldsymbol{w}_g\| \|\boldsymbol{w}_l\|}$ be the angle between a global model's weight $\boldsymbol{w}_g$ and a personalized model's weight $\boldsymbol{w}_l$, $\theta_g$ be the angle between $\boldsymbol{w}_g$ and $\text{Cat}(\mathbf{0}, \boldsymbol{w}_{g,s})$, and $\theta_l$ be the angle between $\boldsymbol{w}_l$ and $\text{Cat}(\mathbf{0}, \boldsymbol{w}_{l,s})$.

**Loss Disparity** The loss disparity of a model $f$ is defined as: $\mathbb{E}_{\mathcal{D}'}[\ell(f(\boldsymbol{x}), y)] - \mathbb{E}_{\mathcal{D}}[\ell(f(\boldsymbol{x}), y)]$, quantifying the generalization performance of a model across environments.

**Adversarial Perturbation.** The adversarial perturbation $\boldsymbol{\delta}_{f,\epsilon} = \boldsymbol{x}_{\text{adv}} - \boldsymbol{x}$ is generated using a model $f$ with budget $\epsilon$. A common way to generate adversarial examples is solving a maximization problem: $\boldsymbol{x}_{\text{adv}} = \arg\max_{\|\boldsymbol{x}^* - \boldsymbol{x}\| \leq \epsilon} \ell(f(\boldsymbol{x}^*), y)$, which returns a data sample that maximizes the model's loss and likely

---

[1]Our analysis could be generalized to more than two environments in a straightforward way.

causes a misprediction. Plugging the definition of $\boldsymbol{\delta}_{f,\epsilon}$ into $\boldsymbol{x}_{\mathrm{adv}} = \arg\max_{\|\boldsymbol{x}^* - \boldsymbol{x}\| \leq \epsilon} \ell(f(\boldsymbol{x}^*), y)$, we have $\boldsymbol{\delta}_{f,\epsilon} = \arg\max_{\|\boldsymbol{\delta}\| \leq \epsilon} \ell(f(\boldsymbol{x} + \boldsymbol{\delta}), y)$. With a small budget $\epsilon$, we can approximate the loss function $\ell$ using the first-order gradient (Miyato et al., 2018; Liang et al., 2021):

$$\boldsymbol{\delta}_{f,\epsilon} = \arg\max_{\|\boldsymbol{\delta}\| \leq \epsilon} \nabla_{\boldsymbol{x}} \ell(f(\boldsymbol{x}), y)^\top \boldsymbol{\delta} = \epsilon \cdot \frac{\nabla_{\boldsymbol{x}} \ell(f(\boldsymbol{x}), y)}{\|\nabla_{\boldsymbol{x}} \ell(f(\boldsymbol{x}), y)\|}. \tag{2}$$

**Adversarial Transferability Measure.** With the adversarial perturbation, we define the adversarial transferability measure that positively correlated with the adversarial transferability:

$$\ell_{g \to l}(f_g, f_l, \boldsymbol{x}, y) = \Big(\ell(f_l(\boldsymbol{x} + \boldsymbol{\delta}_{f_g,\epsilon}), y) - \ell(f_l(\boldsymbol{x}), y)\Big) - \Big(\ell(f_g(\boldsymbol{x} + \boldsymbol{\delta}_{f_g,\epsilon}), y) - \ell(f_g(\boldsymbol{x}), y)\Big), \tag{3}$$

measuring how effective the adversarial perturbation $\boldsymbol{\delta}_{f_g,\epsilon}$ generated by the global model $f_g$ is after applying to the personalized model $f_l$. Specifically, if the adversarial perturbation $\boldsymbol{\delta}_{f_g,\epsilon}$ is less effective on the personalized model $f_l$ due to decreased adversarial transferability, we have $\ell_{g \to l}(f_g, f_l, \boldsymbol{x}, y) < 0$. Otherwise, $\ell_{g \to l}(f_g, f_l, \boldsymbol{x}, y) \geq 0$. For a logistic regression model, its first-order gradient at $\boldsymbol{x}$ is $\Big(\sigma(\boldsymbol{w}^\top \boldsymbol{x}) - y\Big)\boldsymbol{w}$. Here, we can see that the gradient direction depends on the weight $\boldsymbol{w}$ at a given data sample. This allows us to connect the adversarial transferability measure to entanglement using the angles between weights.

### 4.3 An Upper Bound of the Loss Disparity

**Disparity Upper Bound.** To show the necessity of reducing the entanglement to the spurious feature, we connect the entanglement to the spurious feature and the loss disparity, an empirical metric that quantifies a model's generalization performance across environments. The following theorem suggests that the loss disparity of a model is upper bounded by its entanglement to spurious features, quantified by the angle $\theta$ between $\boldsymbol{w}$ and $\mathrm{Cat}(\boldsymbol{0}, \boldsymbol{w}_s)$.

**Theorem 1.** *Under the setting in Section 4.1, for a model $f$ parameterized by $\boldsymbol{w}$, assume the composition $\ell \circ \sigma$ of the activation function $\sigma$ and the loss function $\ell$ is $\rho$-Lipschitz, let $\theta$ be the angle between $\boldsymbol{w}$ and $\mathrm{Cat}(\boldsymbol{0}, \boldsymbol{w}_s)$, we have:*

$$|\mathbb{E}_{\mathcal{D}}[\ell(f(\boldsymbol{x}), y)] - \mathbb{E}_{\mathcal{D}'}[\ell(f(\boldsymbol{x}), y)]| \leq \rho \|\boldsymbol{w}_s\|_2 \cdot W_1(\mathcal{D}_s, \mathcal{D}'_s) = \rho \|\boldsymbol{w}\|_2 \cos\theta \cdot W_1(\mathcal{D}_s, \mathcal{D}'_s). \tag{4}$$

Note that $\cos\theta \in [0, 1]$ by its definition. Theorem 1 implies that the loss disparity is upper bounded by the entanglement measure $\|\boldsymbol{w}\|_2 \cos\theta$ (Section 4.2), the Lipschitz constant $\rho$, and the Wasserstein-1 distance $W_1(\mathcal{D}_s, \mathcal{D}'_s)$. Such a result is intuitive: for a given distribution shift on spurious features $\boldsymbol{x}_s$ that can be quantified by the Wasserstein-1 distance $W_1(\mathcal{D}_s, \mathcal{D}'_s)$, its impact on the loss disparity can be further amplified by up to the product of the entanglement measure $\|\boldsymbol{w}_s\| = \|\boldsymbol{w}\| \cos\theta$ and the Lipschitz constant of $\ell \circ \sigma$.

**Disparity Deviation Upper Bound** Following the result in Theorem 1, we further present an upper bound on the disparity deviation between the global and local models. Our upper bound includes the three angles that are defined in Section 4.2: (1) $\theta_g$, (2) $\theta_l$, and (3) $\theta'$, where the first two angles measure how much the global and personalized models entangle the spurious feature and the angle $\theta'$ quantifies the angle difference between $\boldsymbol{w}_g$ and $\boldsymbol{w}_l$.

**Corollary 1.** *Under the setting in Section 4.1 and the definitions in Section 4.2, with Theorem 4, we have:*

$$\Big| |\mathbb{E}_{\mathcal{D}}[\ell(f_g(\boldsymbol{x}), y)] - \mathbb{E}_{\mathcal{D}'}[\ell(f_g(\boldsymbol{x}), y)]| - |\mathbb{E}_{\mathcal{D}}[\ell(f_l(\boldsymbol{x}), y)] - \mathbb{E}_{\mathcal{D}'}[\ell(f_l(\boldsymbol{x}), y)]| \Big|$$
$$\leq \rho \cdot W_1(\mathcal{D}_s, \mathcal{D}'_s)\Big(\|\boldsymbol{w}_g\| \cos\theta_g + \|\boldsymbol{w}_l\| \cos\theta_l\Big) \tag{5}$$

*In addition, if $\theta' \leq \theta_g$, we have*

$$\Big| |\mathbb{E}_{\mathcal{D}}[\ell(f_g(\boldsymbol{x}), y)] - \mathbb{E}_{\mathcal{D}'}[\ell(f_g(\boldsymbol{x}), y)]| - |\mathbb{E}_{\mathcal{D}}[\ell(f_l(\boldsymbol{x}), y)] - \mathbb{E}_{\mathcal{D}'}[\ell(f_l(\boldsymbol{x}), y)]| \Big|$$
$$\leq \rho \cdot W_1(\mathcal{D}_s, \mathcal{D}'_s)\Big(\|\boldsymbol{w}_g\| \cos\theta_g + \|\boldsymbol{w}_l\| \cos(\theta_g - \theta')\Big) \tag{6}$$

*Otherwise, if $\theta' > \theta_g$, we have:*

$$\left| |\mathbb{E}_{\mathcal{D}}[\ell(f_g(\boldsymbol{x}), y)] - \mathbb{E}_{\mathcal{D}'}[\ell(f_g(\boldsymbol{x}), y)]| - |\mathbb{E}_{\mathcal{D}}[\ell(f_l(\boldsymbol{x}), y)] - \mathbb{E}_{\mathcal{D}'}[\ell(f_l(\boldsymbol{x}), y)]| \right|$$
$$\leq \rho \cdot W_1(\mathcal{D}_s, \mathcal{D}'_s)\left( \|\boldsymbol{w}_g\| \cos\theta_g + \|\boldsymbol{w}_l\| \right) \tag{7}$$

Corollary 1 suggests that the disparity deviation between a pair of models is bounded by the difference between their entanglement measures, where each is composed of an angle term and a norm term. The intuition behind Equations equation 6 and equation 7 is straightforward: in a higher dimensional setting (e.g., $\dim(\boldsymbol{x}_r) \geq 2$ in Section 4.1), the angle $\theta_l$ that quantifies the entanglement of the local model $f_l$ to the spurious feature $\boldsymbol{x}_s$ falls within an interval $[\theta_g - \theta', \theta_g + \theta']$. If $\theta' > \theta_g$, the angle $\theta_l$ is possibly 0 all the time, entangling the local model $f_l$ to the spurious feature. Furthermore, we present a directed analysis of the loss disparity of a local model.

**Corollary 2.** *Under the setting in Section 4.1 and the definitions in Section 4.2, with Theorem 1, if $\theta' \leq \theta_g$, we have*

$$|\mathbb{E}_{\mathcal{D}}[\ell(f_l(\boldsymbol{x}), y)] - \mathbb{E}_{\mathcal{D}'}[\ell(f_l(\boldsymbol{x}), y)]| \leq \rho\|\boldsymbol{w}_{l,s}\| \cdot W_1(\mathcal{D}_s, \mathcal{D}'_s)$$
$$= \rho\|\boldsymbol{w}_l\| \cos\theta_l \cdot W_1(\mathcal{D}_s, \mathcal{D}'_s) \tag{8}$$
$$\leq \rho\|\boldsymbol{w}_l\| \cos(\theta_g - \theta') \cdot W_1(\mathcal{D}_s, \mathcal{D}'_s).$$

*Otherwise, if $\theta' > \theta_g$, we have:*

$$|\mathbb{E}_{\mathcal{D}}[\ell(f_l(\boldsymbol{x}), y)] - \mathbb{E}_{\mathcal{D}'}[\ell(f_l(\boldsymbol{x}), y)]| \leq \rho\|\boldsymbol{w}_l\| \cdot W_1(\mathcal{D}_s, \mathcal{D}'_s). \tag{9}$$

Corollary 2 shows that the loss disparity upper bound of a local model $f_l$ can be upper bounded by its weight norm $\boldsymbol{w}_l$ and the angle difference $\theta'$ between the global and personalized models. Next, we further investigate the relationship between the adversarial transferability between the global and personalized models and the angle $\theta'$ between $\boldsymbol{w}_g$ and $\boldsymbol{w}_l$.

**Adversarial Transferability.** Now, we investigate the connection between adversarial transferability and the loss disparity as well as its deviation. The following theorem bridges the connection between the adversarial transferability measure $\mathbb{E}_{\mathcal{D}}[\ell_{g \to l}(f_g, f_l, \boldsymbol{x}, y)]$ and the angle difference $\theta'$ between the global and personalized models in Corollaries 1 and 2. We show that a high adversarial transferability measure indicates a small $\theta'$ that further implies a low disparity deviation (Corollary 1).

**Theorem 2.** *Under the setting in Section 4.1, let $\epsilon$ be the attack budget, and $\theta'$ be the angle between $\boldsymbol{w}_g$ and $\boldsymbol{w}_l$ we have:*

$$\theta' = \arccos\left( \frac{1}{\epsilon \cdot \bigcirc} \cdot \left[ \epsilon \cdot \square + \diamond + \mathbb{E}_{\mathcal{D}}[\ell_{g \to l}(f_g, f_l, \boldsymbol{x}, y)] \right] \right), \tag{10}$$

*where $\bigcirc = \mathbb{E}_{\mathcal{D}}[\|\nabla_{\boldsymbol{x}}\ell(f_l(\boldsymbol{x}), y)\|]$, $\square = \epsilon\mathbb{E}_{\mathcal{D}}[\|\nabla_{\boldsymbol{x}}\ell(f_g(\boldsymbol{x}), y)\|]$, and $\diamond = \mathbb{E}_{\mathcal{D}}[R_{g,2}(\boldsymbol{x}_{\mathrm{adv}}) - R_{l,2}(\boldsymbol{x}_{\mathrm{adv}})]$ is the expected difference of second-order Taylor remainders between $f_g$ and $f_l$.*

Theorem 2 connects the adversarial transferability measure $\mathbb{E}_{\mathcal{D}}[\ell_{g \to l}(f_g, f_l, \boldsymbol{x}, y)]$ and the angle $\theta'$, which is a key factor in the loss disparity of a personalized model (Corollaries 1 and 2). Such a connection gives rise to an opportunity of minimizing the adversarial transferability measure for reducing the loss disparity of a personalized model. However, in addition to the adversarial transferability measure, Theorem 2 also includes a gradient norm term $\bigcirc$ from the personalized model, a constant $\square$, and a second-order Taylor remainder term $\diamond$, which may need additional treatments.

## 4.4 Necessary Condition

Since the connection between the adversarial transferability measure and the entanglement deviation is angle-based (i.e., $\theta'$), the norm terms (e.g., $\bigcirc = \mathbb{E}_{\mathcal{D}}[\|\nabla_{\boldsymbol{x}}\ell(f_l(\boldsymbol{x}), y)\|]$) may corrupt such a connection and

cause the sub-optimal result in Section 3.3. For example, increasing the gradient norm term $\bigcirc$ of the personalized model $f_l$ may increase $\theta'$ with a fixed adversarial transferability. Also, directly increasing the weight norm $\|\boldsymbol{w}_l\|$ can increase the entanglement deviation even if $\theta' = 0$. Therefore, we need to stabilize the gradient norm term $\bigcirc$ and the weight norm $\|\boldsymbol{w}_l\|$ throughout the personalization step to make the angle-based connection helpful. The following section lists two practices for maintaining adversarial transferability while stabilizing the aforementioned norm terms.

## 5 Methods

In Section 4, Theorem 1 suggests that the accuracy disparity is upper bounded by the Lipschitz constant, our entanglement measure, and the distribution shift measured in Wasserstain-1 distance. Then, in Corollary 1, we show that the disparity deviation between a pair of models is bounded by the difference between their entanglement measures. Further, Theorem 2 shows that maintaining the adversarial transferability can help reduce the difference between the angle terms in the entanglement measures – thus illustrating how we can limit the disparity deviation by maintaining adversarial transferability.

However, such a connection may break if the weight norm of local models significantly increases or decreases during personalization (Section 4.4). To this end, we present an improved approach to maintaining the adversarial transferability while stabilizing the norm terms. Concretely, we first introduce a regularizer to encourage the global and personalized models to make consistent predictions on adversarial examples, maintaining the adversarial transferability. In addition, we add another $L_2$ regularization term to align the weight of the global and personalized models because similar weights lead to similar norm terms. Appending the two regularizers to the loss function during personalization mitigates the increasing accuracy disparity.

### 5.1 Maintaining Adversarial Transferability

**Generating Adversarial Examples.** We employ the same projected gradient descent (PGD) attack (Madry et al., 2018) as Section 3.2, which iteratively perturbs the input using the model's first-order gradient. At iteration $t + 1$, the adversarial example under budget $\epsilon$ is: $\boldsymbol{x}_{\mathrm{adv}}^{t+1} = \mathrm{Proj}_{\|\boldsymbol{x}_{\mathrm{adv}}-\boldsymbol{x}\| \leq \epsilon}(\boldsymbol{x}_{\mathrm{adv}}^t + \alpha \mathrm{sign}(\nabla_{\boldsymbol{x}_{\mathrm{adv}}^t} \ell(f_g(\boldsymbol{x}_{\mathrm{adv}}^t), y)))$, where Proj is a projection operator and $\alpha$ is the attack step size.

**Enforcing Consistent Predictions.** Both the global model $f_g$ and the personalized model $f_l$ take the adversarial example $\boldsymbol{x}_{\mathrm{adv}}$ as input and output $\boldsymbol{z}_g$ and $\boldsymbol{z}_l$ from their last layers, respectively. We increase the adversarial transferability by adding the following regularization term, which minimizes the cross-entropy between $\boldsymbol{z}_g$ and $\boldsymbol{z}_l$ after softmax normalization. Since the global model $f_g$ is fixed as a reference in the personalization step, and its low accuracy disparity is desirable, we use $\boldsymbol{z}_g$ as the ground truth. With $K$ classes, we have:

$$R_{\mathrm{adv}}(\boldsymbol{z}_g, \boldsymbol{z}_l) = \sum_{i=1}^{K} \frac{e^{\boldsymbol{z}_{g,i}}}{\sum_{i=1}^{K} e^{\boldsymbol{z}_{g,i}}} \cdot \log\left(\frac{e^{\boldsymbol{z}_{l,i}}}{\sum_{i=1}^{K} e^{\boldsymbol{z}_{l,i}}}\right).$$

The local model has access to the global model, so there is no additional communication overhead for implementing this regularization. The adversarial examples are computed using the global model once and for all. The computation only needs a few back-propagation, much less than training the global model. Suppose there are $N$ data samples, and the personalization needs $E$ epochs, our approach only generates $N$ adversarial examples once using the global model. In contrast, the standard adversarial training (Madry et al., 2018) generates $N \times E$ adversarial examples.

### 5.2 Aligning Weights and Norms

In addition, we employ a simple and effective strategy by adding an $L_2$ regularization term to the loss function, as a means to minimize the interference from norm terms as is discussed in Section 4.4:

$$R_{L_2}(\boldsymbol{w}_g, \boldsymbol{w}_l) = \|\boldsymbol{w}_g - \boldsymbol{w}_l\|. \tag{11}$$

Table 2: Worst-case accuracy Acc$_{\text{worst}}$ and accuracy disparity Acc$_{\text{disp}}$ across environments of personalized models. Our proposed method improves the worst-environment accuracy and mitigates the accuracy disparity during personalization.

| Method | MNIST | | Coil20 | | CelebA | | BAR | |
|---|---|---|---|---|---|---|---|---|
| | Acc$_{\text{worst}}$ | Acc$_{\text{disp}}$ | Acc$_{\text{worst}}$ | Acc$_{\text{disp}}$ | Acc$_{\text{worst}}$ | Acc$_{\text{disp}}$ | Acc$_{\text{worst}}$ | Acc$_{\text{disp}}$ |
| Global | .847 $\pm$ 6e-4 | .005 $\pm$ 6e-4 | .888 $\pm$ 8e-4 | .001 $\pm$ 1e-3 | .906 $\pm$ 6e-5 | .002 $\pm$ 5e-5 | .720 $\pm$ 4e-5 | .002 $\pm$ 1e-4 |
| FT | .704 $\pm$ 3e-4 | .285 $\pm$ 3e-4 | .804 $\pm$ 4e-3 | .156 $\pm$ 4e-3 | .849 $\pm$ 1e-3 | .114 $\pm$ 1e-3 | .620 $\pm$ 8e-5 | .180 $\pm$ 2e-4 |
| Ditto | .724 $\pm$ 1e-3 | .258 $\pm$ 1e-3 | .952 $\pm$ 2e-4 | .031 $\pm$ 2e-4 | .884 $\pm$ 2e-4 | .082 $\pm$ 2e-4 | .644 $\pm$ 2e-6 | .151 $\pm$ 2e-6 |
| UW | .823 $\pm$ 7e-4 | .145 $\pm$ 2e-5 | .894 $\pm$ 4e-4 | **.021** $\pm$ **4e-4** | .895 $\pm$ 3e-4 | .041 $\pm$ 4e-4 | N/A | N/A |
| JTT | .707 $\pm$ 6e-5 | .278 $\pm$ 6e-5 | .836 $\pm$ 1e-3 | .060 $\pm$ 1e-3 | .836 $\pm$ 2e-4 | .120 $\pm$ 2e-4 | N/A | N/A |
| FedPAC | .755 $\pm$ 5e-4 | .236 $\pm$ 1e-3 | .859 $\pm$ 1e-3 | .102 $\pm$ 2e-3 | .873 $\pm$ 1e-3 | .082 $\pm$ 2e-3 | .654 $\pm$ 3e-3 | .122 $\pm$ 1e-3 |
| FEDORA | .721 $\pm$ 2e-3 | .269 $\pm$ 2e-3 | .847 $\pm$ 1e-3 | .115 $\pm$ 1e-3 | .851 $\pm$ 2e-3 | .109 $\pm$ 3e-3 | .643 $\pm$ 1e-3 | .146 $\pm$ 1e-3 |
| Ours | **.870** $\pm$ **8e-4** | **.081** $\pm$ **2e-5** | **.963** $\pm$ **5e-5** | .023 $\pm$ 1e-5 | **.925** $\pm$ **2e-5** | **.002** $\pm$ **8e-5** | **.730** $\pm$ **2e-4** | **.031** $\pm$ **2e-4** |

Each experiment is repeated 9 times with 3 random seeds for the federated learning step and 3 random seeds for the personalization step.

The motivation behind the $L_2$ term is straightforward: if two models have similar weights, they have similar weight norms and gradient norms, as is shown in the following proposition.

**Proposition 1.** *Under the settings in Section 4.1, assume the composition $\ell \circ \sigma$ of the activation function $\sigma$ and the loss function $\ell$ is $\rho$-Lipschitz, we have:*

$$|\|\boldsymbol{w}_l\| - \|\boldsymbol{w}_g\|| \leq \|\boldsymbol{w}_l - \boldsymbol{w}_g\| \tag{12}$$

*and*

$$|\mathbb{E}_{\mathcal{D}}[\|\nabla_{\boldsymbol{w}_l}\ell(f_l(\boldsymbol{x}), y)\|] - \mathbb{E}_{\mathcal{D}}[\|\nabla_{\boldsymbol{w}_g}\ell(f_g(\boldsymbol{x}), y)\|]| \leq \rho\mathbb{E}_{\mathcal{D}}[\|\boldsymbol{x}\|]\|\boldsymbol{w}_l - \boldsymbol{w}_g\|. \tag{13}$$

We empirically verify that our $L_2$ regularizer reduces the weight norm deviation in Appendix E. Although prior works (Li et al., 2020; T. Dinh et al., 2020; Li et al., 2021) have explored similar regularization methods, we develop the regularization term from a different perspective.

## 6 Experiments

This section presents our experimental results under a federated learning setup with four datasets: MNIST (Deng, 2012), Coil20 (Nene et al., 1996), CelebA (Liu et al., 2015; Caldas et al., 2018), and BAR (Nam et al., 2020), demonstrating that our method mitigates the accuracy disparity. We also show that the benefit of enhanced average accuracy from fine-tuning is preserved. Appendix E further presents a simulation with Gaussian data, experimental results with imperfect global models, and an evaluation of our approach against adversarial attacks.

### 6.1 Setup

**Spurious Features** The MNIST and Coil20 datasets employ a synthetic spurious feature where the object color correlates with the label. Such a synthetic spurious feature allows precise intervention on the spurious attribute of each data sample without interfering with the robust features, matching the setting in Section 4.1. The spurious feature in the BAR dataset is the action background (e.g., rocky cliff). The hair color is a spurious feature that correlates with the gender label in the CelebA dataset.

**Environments** We consider a two-environment setup. Each user gets a different set of samples (Caldas et al., 2018), allocates the samples to the two environments, and randomly picks one environment as the training environment. Combining the training environments eliminates the synthetic spurious correlation in MNIST and Coil20 datasets. The realistic spurious correlation in the BAR and CelebA datasets can diminish but does not necessarily disappear in the global environment.

**Metrics** We consider three metrics in evaluating our approaches: (1) the accuracy disparity $\text{Acc}_{\text{disp}}$ across environments, (2) the worst-case accuracy $\text{Acc}_{\text{worst}}$ among all environments, and (3) the average accuracy in the mixture of all environments (i.e., the global environment). Metrics (1) and (2) measure the generalization performance of personalized models under environment shifts after deployment. Metric (3) quantifies the overall utility of personalized models.

We provide a detailed experimental setup in Appendix D.

## 6.2 Result and Comparison

There are two naive personalization methods: using the global model (Global) and vanilla fine-tuning (FT). In addition, we employ three recent works that are applicable to the federated learning setting (Section 3). Up-weighting (UW) (Sagawa et al., 2020) method is implemented via sampling biased samples and counterexamples with equal probability. Just-train-twice (JTT) (Sagawa et al., 2020) leverages models that have already entangled spurious features (e.g., personalized models with naive fine-tuning) to identify and up-weight counter-examples without explicit annotations. Ditto (Li et al., 2021) adds a regularizer $\lambda\|\boldsymbol{w}_l - \boldsymbol{w}_g\|$ during personalization to limit the deviation of personalized models from the global model. We select models using the validation accuracy minus the adversarial transferability degradation for ours and using the validation accuracy for others.

### 6.2.1 Summary of Results

Tables 2 and 3 show the main results with the two metrics on generalization performance in Section 6.1. Our method reduces the accuracy disparity of personalized models from 18.38% to 3.42% compared to vanilla fine-tuning, outperforming other competitors (Table 2). On average, our method also preserves the enhanced average accuracy from fine-tuning, resulting in 4.48% accuracy improvement in global environments compared to the global model (Table 3).

### 6.2.2 Analysis of Results

One limitation of up-weighting-based methods (i.e., UW and JTT) relates to the diversity of counter-examples, which do not agree with spurious correlations. Counter-examples are few in datasets, leading to low diversity.

To explore the impact of counter-examples' diversity on disentangling spurious features from a model, we vary the diversity of counter-examples and adjust the up-weighting factors accordingly. Specifically, we sample a factor of 0.02, 0.025, 0.033, 0.05, and 0.1 biased samples from the MNIST dataset and re-color them to become counter-examples. The factor in the sampling step is called the "sampling factor". Then, we up-weight the counter-examples by a factor of 50, 40, 30, 20, and 10, respectively, keeping the total number of counter-examples consistent. Here, the counter-examples have less diversity if generated by a small number of biased data samples with a large up-weighting factor. Experimental results in Figure 2 show that, as the diversity reduces, the accuracy disparity of personalized models increases, supporting our hypothesis on the sample diversity. Therefore, our method is applicable in the scarcity of counter-examples while up-weighting-based methods fail.

The issue with Ditto is that greedily maximizing the accuracy in training environments often leads to small $\lambda$, giving personalized models spaces to entangle spurious features. FedPAC (Xu et al., 2023) and FEDORA (Wu et al., 2023), which promote knowledge transfer between personalized models on clients that are similar to each other, can exacerbate the entanglement to spurious features because they encourage the knowledge transfer between clients that share similar spurious features while isolating clients from different environments.

## 6.3 Ablation Study

We conduct an ablation study on the MNIST dataset and plot the accuracy and loss curves, which are shown to have similar behaviors. Figures 3b and 3f demonstrate that only enforcing consistent predictions on adversarial examples is insufficient. Figure 3c and 3g further show that applying the $L_2$ regularization term alone does not address the accuracy or loss disparity. Aligning weights of the global and personalized models

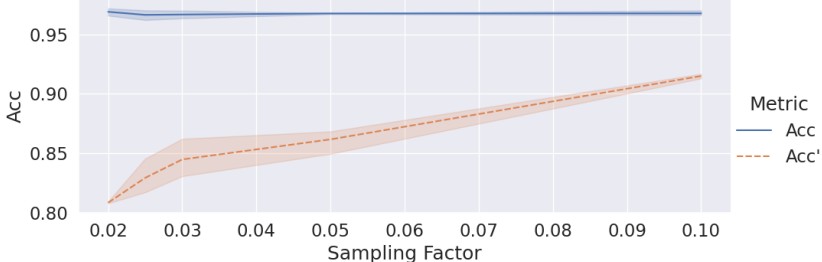

Figure 2: The up-weighting method is less effective with few counter-examples, resulting in a large accuracy disparity of personalized model in the training environment (Acc) and the global environment (Acc′), if the counter-examples are generated by a small number of biased data samples using a small sampling factor and a large up-weighting factor. The up-weighting factor is set to be the reciprocal of the sampling factor.

Table 3: Average accuracy personalized models over different environments. Our proposed method preserves the enhanced accuracy of personalization.

| Method | MNIST | Coil20 | CelebA | BAR |
|---|---|---|---|---|
| Global | $.849 _{\pm \text{ 6e-4}}$ | $.888 _{\pm \text{ 1e-3}}$ | $.919 _{\pm \text{ 5e-5}}$ | $.721 _{\pm \text{ 1e-4}}$ |
| FT | $.846 _{\pm \text{ 3e-4}}$ | $.882 _{\pm \text{ 4e-3}}$ | $.906 _{\pm \text{ 1e-3}}$ | $.710 _{\pm \text{ 2e-4}}$ |
| Ditto | $.853 _{\pm \text{ 1e-3}}$ | $.968 _{\pm \text{ 2e-4}}$ | $.925 _{\pm \text{ 2e-4}}$ | $.719 _{\pm \text{ 2e-6}}$ |
| FedPAC | $.873 _{\pm \text{ 1e-3}}$ | $.910 _{\pm \text{ 2e-3}}$ | $.914 _{\pm \text{ 2e-3}}$ | $.715 _{\pm \text{ 3e-3}}$ |
| FEDORA | $.855 _{\pm \text{ 2e-3}}$ | $.904 _{\pm \text{ 1e-3}}$ | $.905 _{\pm \text{ 3e-3}}$ | $.716 _{\pm \text{ 1e-3}}$ |
| Ours | $\mathbf{.911} _{\pm \text{ 2e-5}}$ | $\mathbf{.974} _{\pm \text{ 1e-5}}$ | $\mathbf{.926} _{\pm \text{ 8e-5}}$ | $\mathbf{.745} _{\pm \text{ 2e-4}}$ |

by applying the $L_2$ regularization term while maintaining the adversarial transferability address the accuracy and loss disparity as Figures 3d and 3h show, respectively. Both components in our methods are helpful, and a combination of them achieves the best results.

We also explored enforcing the global and personalized models to make consistent predictions on benign samples. However, the accuracy disparity of personalized models increases by 9.66% (19.25% relative increase) with fine-tuning 16 batches. We hypothesize that personalized models could use a combination of robust and spurious features to make the same prediction as the global model on benign samples, which only uses robust features and is disentangled from spurious features.

### 6.4 Visualizing Gradient Magnitudes

We plot the distribution of the ratio $\frac{\|\nabla_{\boldsymbol{x}} \ell(f_l(\boldsymbol{x}), y)\|}{\|\nabla_{\boldsymbol{x}} \ell(f_g(\boldsymbol{x}), y)\|}$ across data samples with 16 batches of fine-tuning on the MNIST dataset, using the global model as a reference. Figure 4 suggests enforcing consistent predictions using logits (our approach) achieves lower gradient magnitude deviation than maintaining the adversarial transferability using flipped labels (naive approach). The $L_2$ regularizer also stabilizes gradient magnitudes.

## 7 Conclusion and Future Work

In this work, we show a risk of prior federated learning personalization methods with spurious features, which lead to high accuracy disparity between environments. Then, we develop a strategy to reduce the accuracy disparity by maintaining the adversarial transferability between the global and personalized models. Both empirical and theoretical results show that our strategy is effective. Extensions of this work include incorporating text data, which often requires different methods for generating adversarial examples.

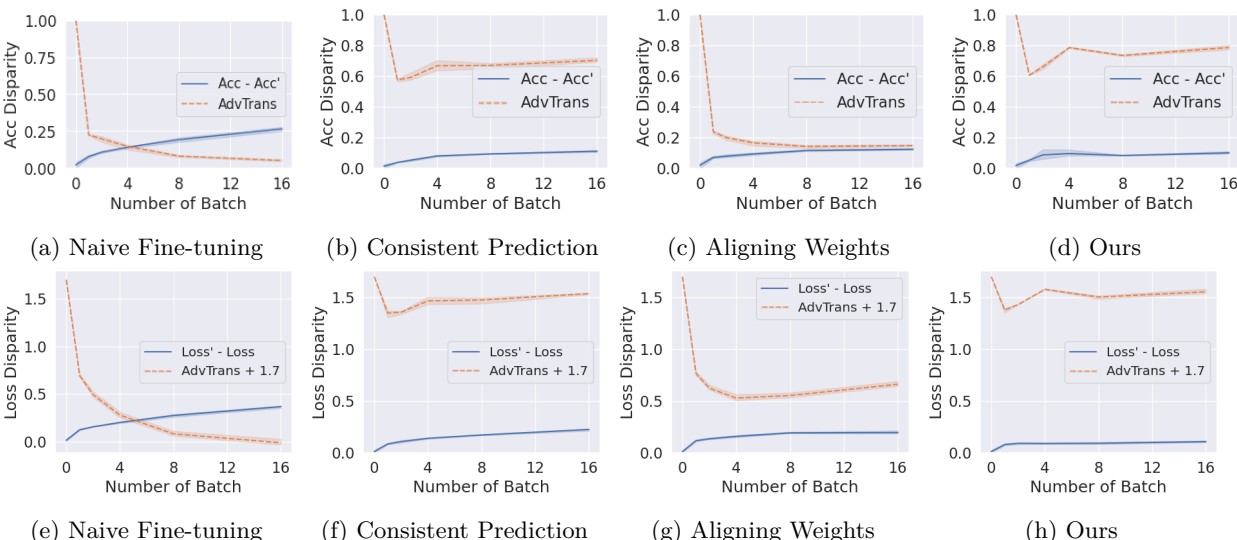

Figure 3: The accuracy disparity (Acc - Acc′) and the loss disparity (Loss′ - Loss) between the training and the global environments. These two disparities correlate with the adversarial transferability measure (AdvTrans) differently with different methods. We add 1.7 to the adversarial transferability measure over the loss to ease reading. Accuracy and loss disparities have similar behaviors. Combining the two practices in our approach addresses the accuracy and loss disparities.

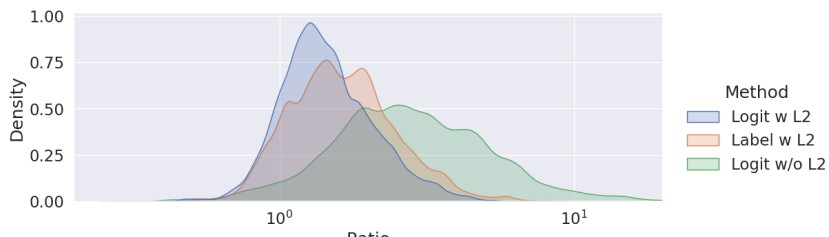

Figure 4: Distribution of the ratio $\frac{\|\nabla_{\boldsymbol{x}}\ell(f_l(\boldsymbol{x}),y)\|}{\|\nabla_{\boldsymbol{x}}\ell(f_g(\boldsymbol{x}),y)\|}$. Using our approach and enforcing consistent predictions with logits achieves lower gradient magnitude deviation than adding adversarial examples with flipped labels to personalization. The $L_2$ regularizer also stabilizes gradient magnitudes.

# 8 Acknowledgement

XW and KN are partially supported by a research grant from the C3.ai Digital Transformation Institute (DTI). This work of SK is partially supported by NSF III 2046795, IIS 1909577, CCF 1934986, NIH 1R01MH116226-01A, NIFA award 2020-67021-32799, the Alfred P. Sloan Foundation, and Google Inc. The work of HZ is partially supported by the Defense Advanced Research Projects Agency (DARPA) under Cooperative Agreement Number: HR00112320012 and a research grant from the Amazon-Illinois Center on AI for Interactive Conversational Experiences (AICE).

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

# A    Spurious Feature Examples

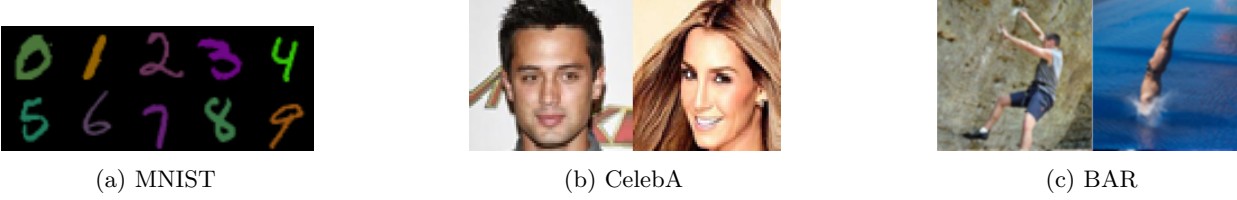

(a) MNIST                    (b) CelebA                    (c) BAR

Figure 5: Datasets with spurious features. The object color spuriously correlates with the digit in MNIST (a). The hair color spuriously correlates with the gender in the CelebA dataset (b). The background spuriously correlates with the action in the BAR dataset (c).

# B    Notation

Table 4: Table of Notation

| Symbol | Description |
| --- | --- |
| $\boldsymbol{x}, y$ | A pair of data sample and label |
| $\mathrm{Cat}(\cdot, \cdot)$ | A concatenation of two vectors |
| $\boldsymbol{x}_r, \boldsymbol{x}_s$ | The robust features and spurious features in $\boldsymbol{x} = \mathrm{Cat}(\boldsymbol{x}_r, \boldsymbol{x}_s)$, respectively |
| $f_g$ | The global model |
| $f_l$ | The personalized local model |
| $\boldsymbol{\delta}_{f_g,\epsilon}$ | An adversarial perturbation generated by the global model $f_g$ with attack budget $\epsilon$ |
| $\mathcal{D}$ | A distribution in the training environment |
| $\mathcal{D}'$ | A distribution in the other environment that is not accessible during training |
| $W_1(\cdot, \cdot)$ | Wasserstain-1 distance between two distributions |

# C    Proofs

**Theorem 1.** *Under the setting in Section 4.1, for a model $f$ parameterized by $\boldsymbol{w}$, assume the composition $\ell \circ \sigma$ of the activation function $\sigma$ and the loss function $\ell$ is $\rho$-Lipschitz, let $\theta$ be the angle between $\boldsymbol{w}$ and $\mathrm{Cat}(\boldsymbol{0}, \boldsymbol{w}_s)$, we have:*

$$|\mathbb{E}_{\mathcal{D}}[\ell(f(\boldsymbol{x}), y)] - \mathbb{E}_{\mathcal{D}'}[\ell(f(\boldsymbol{x}), y)]| \le \rho \|\boldsymbol{w}_s\|_2 \cdot W_1(\mathcal{D}_s, \mathcal{D}'_s) = \rho \|\boldsymbol{w}\|_2 \cos\theta \cdot W_1(\mathcal{D}_s, \mathcal{D}'_s). \tag{14}$$

*Proof.* With the data generating process in Section 4.1, we have:

$$|\mathbb{E}_{\mathcal{D}}[\ell(f(\boldsymbol{x}), y)] - \mathbb{E}_{\mathcal{D}'}[\ell(f(\boldsymbol{x}), y)]| = |\mathbb{E}_{\mathcal{D}, \mathcal{D}'}[\ell(f(\boldsymbol{x}), y) - \ell(f(\boldsymbol{x}'), y)]|. \tag{15}$$

Since the loss function is $\rho$-Lipschitz, we further have:

$$|\mathbb{E}_{\mathcal{D}, \mathcal{D}'}[\ell(f(\boldsymbol{x}), y) - \ell(f(\boldsymbol{x}'), y)]| \le \rho |\mathbb{E}_{\mathcal{D}, \mathcal{D}'}[f(\boldsymbol{x}) - f(\boldsymbol{x}')]|. \tag{16}$$

Under the given data generating process in Section 4.2, we know that $\boldsymbol{x} - \boldsymbol{x}' = \mathrm{Cat}(\boldsymbol{0}, \boldsymbol{x}_s - \boldsymbol{x}'_s)$, yielding:

$$|\mathbb{E}_{\mathcal{D}, \mathcal{D}'}[f(\boldsymbol{x}) - f(\boldsymbol{x}')]| \le \|\boldsymbol{w}_s\|_2 \mathbb{E}_{\mathcal{D}_s, \mathcal{D}'_s}[\boldsymbol{x}_s - \boldsymbol{x}'_s]| = \|\boldsymbol{w}\|_2 \cos\theta |\mathbb{E}_{\mathcal{D}_s, \mathcal{D}'_s}[\boldsymbol{x}_s - \boldsymbol{x}'_s]|. \tag{17}$$

By the definition of Wasserstein-1 distance, we have:

$$|\mathbb{E}_{\mathcal{D}_s, \mathcal{D}'_s}[\boldsymbol{x}_s - \boldsymbol{x}'_s]| \le \mathbb{E}_{\mathcal{D}_s, \mathcal{D}'_s}[|\boldsymbol{x}_s - \boldsymbol{x}'_s|] \le W_1(\mathcal{D}_s, \mathcal{D}'_s). \tag{18}$$

$\square$

**Theorem 2.** *Under the setting in Section 4.1, let $\epsilon$ be the attack budget, and $\theta'$ be the angle between $\boldsymbol{w}_g$ and $\boldsymbol{w}_l$ we have:*

$$\theta' = \arccos\left(\frac{1}{\epsilon \cdot \bigcirc} \cdot \left[\epsilon \cdot \square + \diamond + \mathbb{E}_{\mathcal{D}}[\ell_{g \to l}(f_g, f_l, \boldsymbol{x}, y)]\right]\right), \tag{19}$$

*where $\bigcirc = \mathbb{E}_{\mathcal{D}}[\|\nabla_{\boldsymbol{x}}\ell(f_l(\boldsymbol{x}), y)\|]$, $\square = \epsilon\mathbb{E}_{\mathcal{D}}[\|\nabla_{\boldsymbol{x}}\ell(f_g(\boldsymbol{x}), y)\|]$, and $\diamond = \mathbb{E}_{\mathcal{D}}[R_{g,2}(\boldsymbol{x}_{\mathrm{adv}}) - R_{l,2}(\boldsymbol{x}_{\mathrm{adv}})]$ is the expected difference of second-order Taylor remainders between $f_g$ and $f_l$.*

*Proof.* Expanding $\ell_{g \to l}(f_g, f_l, \boldsymbol{x}, y)$ at $(\boldsymbol{x}, y)$ and recalling the definition of adversarial perturbation $\boldsymbol{\delta}_{f,\epsilon} = \epsilon \cdot \frac{\nabla_{\boldsymbol{x}}\ell(f(\boldsymbol{x}), y)}{\|\nabla_{\boldsymbol{x}}\ell(f(\boldsymbol{x}), y)\|}$ in Equation (2), we have:

$$
\begin{aligned}
&\ell_{g \to l}(f_g, f_l, \boldsymbol{x}, y) \\
&= \left(\ell(f_l(\boldsymbol{x} + \boldsymbol{\delta}_{f_g,\epsilon}), y) - \ell(f_l(\boldsymbol{x}), y)\right) - \left(\ell(f_g(\boldsymbol{x} + \boldsymbol{\delta}_{f_g,\epsilon}), y) - \ell(f_g(\boldsymbol{x}), y)\right) \\
&= \nabla_{\boldsymbol{x}}\ell(f_l(\boldsymbol{x}), y)^{\top}\boldsymbol{\delta}_{f_g,\epsilon} + R_{l,2}(\boldsymbol{x}_{\mathrm{adv}}) - \nabla_{\boldsymbol{x}}\ell(f_g(\boldsymbol{x}), y)^{\top}\boldsymbol{\delta}_{f_g,\epsilon} - R_{g,2}(\boldsymbol{x}_{\mathrm{adv}}) \\
&= \nabla_{\boldsymbol{x}}\ell(f_l(\boldsymbol{x}), y)^{\top} \cdot \epsilon \cdot \frac{\nabla_{\boldsymbol{x}}\ell(f_g(\boldsymbol{x}), y)}{\|\nabla_{\boldsymbol{x}}\ell(f_g(\boldsymbol{x}), y)\|} - \epsilon\|\nabla_{\boldsymbol{x}}\ell(f_g(\boldsymbol{x}), y)\| + R_{l,2}(\boldsymbol{x}_{\mathrm{adv}}) - R_{g,2}(\boldsymbol{x}_{\mathrm{adv}}),
\end{aligned}
\tag{20}
$$

where $\boldsymbol{x}_{\mathrm{adv}} = \boldsymbol{x} + \boldsymbol{\delta}_{f_g,\epsilon}$ and $R_{l,2}(\boldsymbol{x}_{\mathrm{adv}}) - R_{g,2}(\boldsymbol{x}_{\mathrm{adv}})$ is the difference of second-order Taylor remainders between $f_g$ and $f_l$.

For logistic regression models $f_g$ and $f_l$, we have:

$$\nabla_{\boldsymbol{x}}\ell(f_g(\boldsymbol{x}), y) = \left(\sigma(\boldsymbol{w}_g^{\top}\boldsymbol{x}) - y\right)\boldsymbol{w}_g \text{ and } \nabla_{\boldsymbol{x}}\ell(f_l(\boldsymbol{x}), y) = \left(\sigma(\boldsymbol{w}_l^{\top}\boldsymbol{x}) - y\right)\boldsymbol{w}_l. \tag{21}$$

Since the label $y \in \{0, 1\}$ of each data sample $\boldsymbol{x}$ is fixed, and the logits $\sigma(\boldsymbol{w}_g^{\top}\boldsymbol{x})$ and $\sigma(\boldsymbol{w}_l^{\top}\boldsymbol{x})$ are in $(0, 1)$, the product of scalars $\sigma(\boldsymbol{w}_g^{\top}\boldsymbol{x}) - y$ and $\sigma(\boldsymbol{w}_l^{\top}\boldsymbol{x}) - y$ is always positive. This further implies that the angle between gradients $\nabla_{\boldsymbol{x}}\ell(f_g(\boldsymbol{x}), y)$ and $\nabla_{\boldsymbol{x}}\ell(f_l(\boldsymbol{x}), y)$ equals the angle $\theta'$ between weight vectors $\boldsymbol{w}_g$ and $\boldsymbol{w}_l$. In addition, the angle $\theta'$ between weight vectors is consistent across data samples. Therefore, we further have:

$$\nabla_{\boldsymbol{x}}\ell(f_l(\boldsymbol{x}), y)^{\top} \cdot \epsilon \cdot \frac{\nabla_{\boldsymbol{x}}\ell(f_g(\boldsymbol{x}), y)}{\|\nabla_{\boldsymbol{x}}\ell(f_g(\boldsymbol{x}), y)\|} = \epsilon\|\nabla_{\boldsymbol{x}}\ell(f_l(\boldsymbol{x}), y)\|\cos\theta'. \tag{22}$$

Plugging Equation 22 into Equation 20 and taking the expectation over $\mathcal{D}$, we have:

$$
\begin{aligned}
&\mathbb{E}_{\mathcal{D}}[\ell_{g \to l}(f_g, f_l, \boldsymbol{x}, y)] \\
&= \epsilon\mathbb{E}_{\mathcal{D}}[\|\nabla_{\boldsymbol{x}}\ell(f_l(\boldsymbol{x}), y)\|]\cos\theta' - \epsilon\mathbb{E}_{\mathcal{D}}[\|\nabla_{\boldsymbol{x}}\ell(f_g(\boldsymbol{x}), y)\|] + \mathbb{E}_{\mathcal{D}}[R_{l,2}(\boldsymbol{x}_{\mathrm{adv}}) - R_{g,2}(\boldsymbol{x}_{\mathrm{adv}})].
\end{aligned}
\tag{23}
$$

Rearranging the terms in Equation 23, we have:

$$
\begin{aligned}
\cos\theta' &= \frac{\epsilon\mathbb{E}_{\mathcal{D}}[\|\nabla_{\boldsymbol{x}}\ell(f_g(\boldsymbol{x}), y)\|] + \mathbb{E}_{\mathcal{D}}[R_{g,2}(\boldsymbol{x}_{\mathrm{adv}}) - R_{l,2}(\boldsymbol{x}_{\mathrm{adv}})] + \mathbb{E}_{\mathcal{D}}[\ell_{g \to l}(f_g, f_l, \boldsymbol{x}, y)]}{\epsilon\mathbb{E}_{\mathcal{D}}[\|\nabla_{\boldsymbol{x}}\ell(f_l(\boldsymbol{x}), y)\|]} \\
&= \frac{1}{\epsilon \cdot \bigcirc} \cdot \left[\epsilon \cdot \square + \diamond + \mathbb{E}_{\mathcal{D}}[\ell_{g \to l}(f_g, f_l, \boldsymbol{x}, y)]\right].
\end{aligned}
\tag{24}
$$

Then, taking the arccos operator completes the proof. $\qquad\square$

**Proposition 1.** *Under the settings in Section 4.1, assume the composition $\ell \circ \sigma$ of the activation function $\sigma$ and the loss function $\ell$ is $\rho$-Lipschitz, we have:*

$$\|\|\boldsymbol{w}_l\| - \|\boldsymbol{w}_g\|\| \leq \|\boldsymbol{w}_l - \boldsymbol{w}_g\| \tag{25}$$

*and*

$$|\mathbb{E}_{\mathcal{D}}[\|\nabla_{\boldsymbol{w}_l}\ell(f_l(\boldsymbol{x}), y)\|] - \mathbb{E}_{\mathcal{D}}[\|\nabla_{\boldsymbol{w}_g}\ell(f_g(\boldsymbol{x}), y)\|]| \leq \rho\mathbb{E}_{\mathcal{D}}[\|\boldsymbol{x}\|]\|\boldsymbol{w}_l - \boldsymbol{w}_g\|. \tag{26}$$

*Proof.* With the triangle inequality, we have:

$$\|\boldsymbol{w}_l\| = \|\boldsymbol{w}_l + \boldsymbol{w}_g - \boldsymbol{w}_g\| \leq \|\boldsymbol{w}_g\| + \|\boldsymbol{w}_l - \boldsymbol{w}_g\| \tag{27}$$

and

$$\|\boldsymbol{w}_g\| = \|\boldsymbol{w}_g + \boldsymbol{w}_l - \boldsymbol{w}_l\| \leq \|\boldsymbol{w}_l\| + \|\boldsymbol{w}_g - \boldsymbol{w}_l\|. \tag{28}$$

Since $\|\boldsymbol{w}_g - \boldsymbol{w}_l\| = \|\boldsymbol{w}_l - \boldsymbol{w}_g\|$, moving $\|\boldsymbol{w}_g\|$ and $\|\boldsymbol{w}_l\|$ to the left-hand-side in Equations 27 and 28, respectively, we have $|\|\boldsymbol{w}_l\| - \|\boldsymbol{w}_g\|| \leq \|\boldsymbol{w}_l - \boldsymbol{w}_g\|$.

Similarly, we have:

$$\begin{aligned}
\|\nabla_{\boldsymbol{w}_l}\ell(f_l(\boldsymbol{x}),y)\| &= \|\nabla_{\boldsymbol{w}_l}\ell(f_l(\boldsymbol{x}),y) + \nabla_{\boldsymbol{w}_g}\ell(f_g(\boldsymbol{x}),y) - \nabla_{\boldsymbol{w}_g}\ell(f_g(\boldsymbol{x}),y)\| \\
&\leq \|\nabla_{\boldsymbol{w}_g}\ell(f_g(\boldsymbol{x}),y)\| + \|\nabla_{\boldsymbol{w}_l}\ell(f_l(\boldsymbol{x}),y) - \nabla_{\boldsymbol{w}_g}\ell(f_g(\boldsymbol{x}),y)\| \\
&\leq \|\nabla_{\boldsymbol{w}_g}\ell(f_g(\boldsymbol{x}),y)\| + \rho\|\boldsymbol{x}\|\|\boldsymbol{w}_l - \boldsymbol{w}_g\|
\end{aligned} \tag{29}$$

and

$$\|\nabla_{\boldsymbol{w}_g}\ell(f_g(\boldsymbol{x}),y)\| \leq \|\nabla_{\boldsymbol{w}_g}\ell(f_l(\boldsymbol{x}),y)\| + \rho\|\boldsymbol{x}\|\|\boldsymbol{w}_g - \boldsymbol{w}_l\|. \tag{30}$$

Recalling that $\|\boldsymbol{w}_g - \boldsymbol{w}_l\| = \|\boldsymbol{w}_l - \boldsymbol{w}_g\|$, combining Equations 29 and 30 results in:

$$\|\nabla_{\boldsymbol{w}_g}\ell(f_g(\boldsymbol{x}),y)\| - \|\nabla_{\boldsymbol{w}_g}\ell(f_l(\boldsymbol{x}),y)\| \leq |\|\nabla_{\boldsymbol{w}_g}\ell(f_g(\boldsymbol{x}),y)\| - \|\nabla_{\boldsymbol{w}_g}\ell(f_l(\boldsymbol{x}),y)\|| \leq \rho\|\boldsymbol{x}\|\|\boldsymbol{w}_g - \boldsymbol{w}_l\|. \tag{31}$$

Taking the expectation and absolute value over both sides of Equation 31 completes the proof. $\qquad\square$

## D  Detailed Experimental Setting

### D.1  Data Partition

We distribute the MNIST, Coil20, and BAR datasets across 50 clients. Each client has data samples from 5 different classes. These data samples are further grouped into two environments according to their spurious correlations. Each client randomly picks one environment for training. Local datasets are further partitioned to train/validation/test set with a ratio of 72:8:20, following prior work (Li et al., 2021). Combining test sets from the two environments gets test sets in global environments.

For the CelebA dataset, we let each client represent 20 celebrities. One celebrity only appears on one client. The blond hair correlates with the female in the training environment and correlates with the male in the other environment that is not used for training. Note that non-blond hair colors do not correlate with any gender. In the personalization step, we select the clients with more than 5 blond female data samples in the training environment and more than 5 blond male data samples in the other environment that is not used for training. We select these clients because they provide enough samples to create the spurious correlation and evaluate entanglements to the spurious correlation.

### D.2  Hyper-parameters

We use Adam optimizer (Kingma & Ba, 2015) throughout our experiments with a learning rate of 1e-4 for MNIST, CelebA, and BAR and 2e-4 for Coil20. Although stochastic gradient descent (SGD) optimizer is more common in vision-related tasks, the Adam optimizer always leads to lower accuracy disparity. We train the global model for 500 rounds. 5 clients are selected per round, each performing 5 epochs of local updates. We tune the coefficients of the adversarial transferability and $L_2$ regularization terms from $\{0.01, 0.1, 1.0, 10.0\}$ and select the largest value that does not decrease the validation accuracy during penalization. We start the attack budget at 0.031 (i.e., $\frac{8}{255}$) and gradually decrease it such that $30\%-50\%$ of the attack succeeds. A large budget will make the attack too strong and push the adversarial examples far across the decision boundary, making the regularization method less effective. We configure $\epsilon$ to 0.031/0.01/0.031 for MNIST/CelebA/Coil20, respectively. We fine-tune the global model for 5 epochs on MNIST/BAR and 10 epochs on Coil20/CelebA, which are sufficient for the personalized models to converge. The clients with the most data samples fine-tune

the penalized models for a total of 80/40/30 batches on MNIST/CelebA/Coil20 datasets. Note that we may not select the personalized model with the most fine-tuning batches for reporting. In the up-weighting (UW) and just-train-twice (JTT) method, we up-sample the residual by a factor of 50. In Ditto, we tune its $\lambda$ from $\{0.1, 1.0\}$.

### D.3 Neural Network Architecture

We use CNN 28x28 for the MNIST dataset and CNN 64x64 for the CelebA and the Coil20 datasets, as are listed in Table 5. We use a pre-trained ResNet-18 (He et al., 2016) for the BAR dataset.

Table 5: Neural Network Architecture

| CNN 28 × 28 | CNN 64 × 64 |
| --- | --- |
| Input: $\mathbb{R}^{3 \times 28 \times 28}$ | Input: $\mathbb{R}^{3 \times 64 \times 64}$ |
| 4×4 conv, 64 BN LReLU, stride 2 | 4×4 conv, 64 BN LReLU, stride 2 |
| 4×4 conv, 128 BN LReLU, stride 2 | 4×4 conv, 64 BN LReLU, stride 2 |
| FC 4096 ReLU | FC 4096 ReLU |
| FC 10 | FC 10 |

## E More Experimental Results

### E.1 Transferability Correlates with Disparity

The additional empirical studies on Coil20 and CelebA datasets are shown in Figure 6.

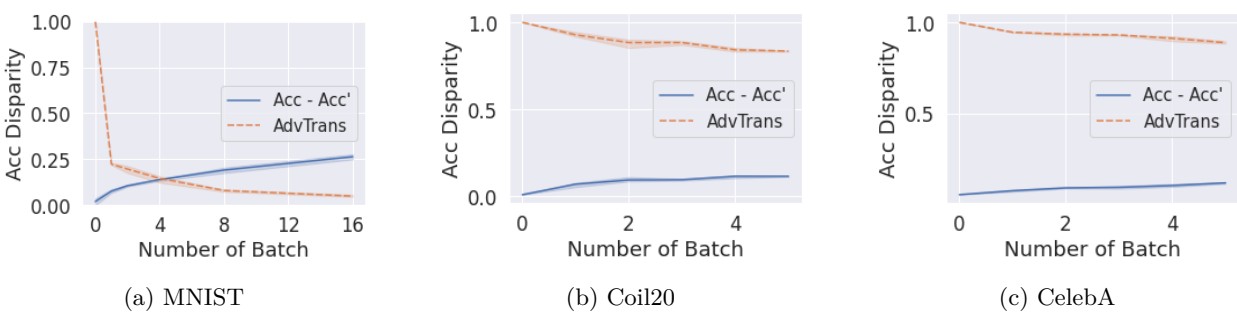

(a) MNIST  (b) Coil20  (c) CelebA

Figure 6: As the personalized models entangle the spurious feature and increase their accuracy disparity (Acc - Acc′), the adversarial transferability (AdvTrans) decreases. Here, the accuracy disparity deviation also increases as the accuracy disparity of personalized models deviates from that of the global model, which is fixed as a reference.

### E.2 Simulations

We construct a synthetic dataset where $y \in [0, 1]$, $\boldsymbol{x} = [\boldsymbol{x}_r, \boldsymbol{x}_s]$ and $\boldsymbol{x}_r \sim \mathcal{N}(2y-1, 1)$ and $\boldsymbol{x}_s \sim \mathcal{N}\big(c(2y-1), 1\big)$ where the $c$ is 1 if the client index mods 2 is 0. Otherwise, the $c$ is -1 (Figures 7a and 7c). In the experiments shown in Figures 7b and 7d, the $c$ is 1 if the client index mods 2 is 0; otherwise, the $c$ is 0. We employ a linear model $y = \boldsymbol{w}^\top \boldsymbol{x}$ where $\boldsymbol{w} = \mathrm{Cat}(\boldsymbol{w}_r, \boldsymbol{w}_s)$.

Our simulation starts with the global model. We use the federated averaging algorithm with 50 clients, each with 128 data samples, and select 5 clients per round. Figure 7 shows that the gradient $\boldsymbol{g}_{\boldsymbol{w}_s}$ of $\boldsymbol{w}_s$ is noisier and closer to 0 than the gradient $\boldsymbol{g}_{\boldsymbol{w}_r}$ of $\boldsymbol{w}_r$. As a result, $\boldsymbol{w}_r$ increases during the training while $\boldsymbol{w}_s$ does not increase or increases much slower, as is shown in Figure 7. These results suggest that when the spurious

correlation varies across clients, the model does not entangle the spurious feature or only slightly entangle the spurious feature.

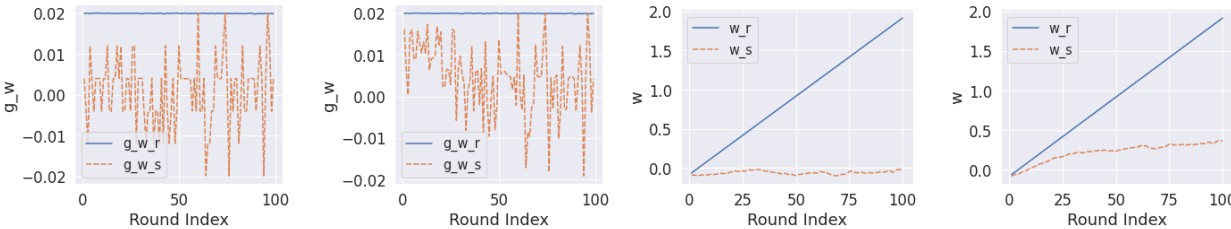

(a)  Spurious  Correlation Shifts Across Users (b) 50% Users Have a Stationary Spurious Correlation (c)  Spurious  Correlation Shifts Across Users (d) 50% Users Have a Stationary Spurious Correlation

Figure 7: The gradients and weights of a linear model during training. The gradient $g_{\boldsymbol{w}_s}$ of $\boldsymbol{w}_s$ is more noisy and closer to 0 than the gradient $g_{\boldsymbol{w}_r}$ of $\boldsymbol{w}_r$. Therefore, $\boldsymbol{w}_s$ stays around 0.

Then, we personalize the global model whose $\boldsymbol{w} = [2.0, 0.0]$ on clients with shifting spurious correlations. Results in Figure 8 suggest that our approach limits $\boldsymbol{w}_s$ around 0 with minimal fluctuation.

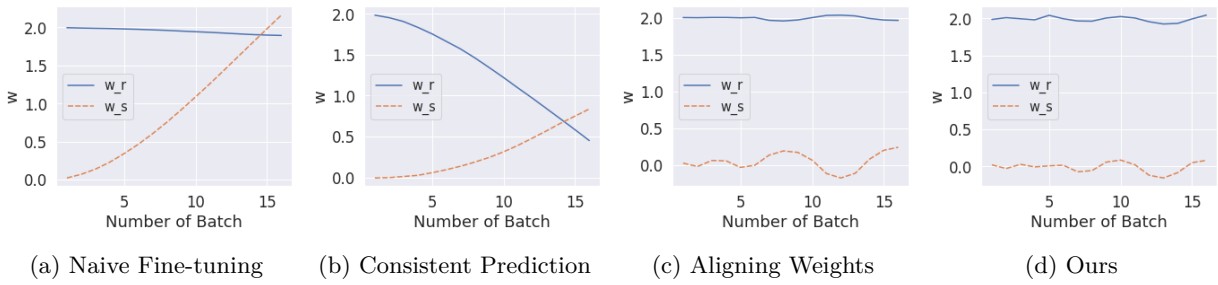

(a) Naive Fine-tuning (b) Consistent Prediction (c) Aligning Weights (d) Ours

Figure 8: Weights of a linear model during fine-tuning. Our approach reduces the co-efficiency $\boldsymbol{w}_s$ of the spurious feature $\boldsymbol{x}_s$ most effectively. Aligning weights also limits $\boldsymbol{w}_s$ but has a more significant fluctuation than ours.

### E.3   Personalization with Imperfect Global Model

Our approach still bounds the disparity deviation of personalized models when the global model already entangles spurious features. Specifically, for two global models with accuracy disparities of 0.5% and 2.1% on MNIST, the accuracy disparity of personalized models is 8.1% and 12.4%, respectively. In contrast, fine-tuning increases both disparities to 28.5%.

### E.4   Robustness to Adversarial Attack

Although we add adversarial examples with flipped labels during personalization, the robustness of personalized models from our approach does not degrade significantly. We evaluate our approach against transferable adversarial examples from the global model and direct adversarial attacks on personalized models with attack configurations in Appendix D. Our approach only increases the success rate of transferable attacks from 56.1% to 68.2% and that of direct attacks from 47.8% to 50.1% on MNIST. Both transferable attacks use the global model to generate adversarial examples. The success rate of transferable attacks is not 100% because personalized models are trained with different benign and adversarial samples locally. Transferable attacks can have higher success rates than direct attacks since only the adversarial examples that "fool" global models are transferred.

