# OpenReview forum: "Personalized Federated Learning with Spurious Features: An Adversarial Approach"
_TMLR — Accepted by TMLR_

### Review · Reviewer_rNGy · 2023-09-18

**Summary Of Contributions:**

This work addresses the challenge of personalizing federated learning models, where fine-tuning global models for local clients can lead to reduced generalization performance due to spurious features at local clients. In this setup, the spurious features correlate with the label in each client's training environment, but the global environment diminishes these correlations. While global federated learning models suffer less from spurious features, local fine-tuning may reintroduce vulnerability to them. To mitigate this, the authors propose a novel strategy that maintains adversarial transferability between global and personalized models. Empirical results on object and action recognition tasks demonstrate that this approach prevents personalized models from exploiting spurious features while retaining the benefits of accuracy enhancement through fine-tuning.

**Audience:**

Yes

**Broader Impact Concerns:**

No ethical concerns have identified for this paper.

**Claims And Evidence:**

Yes

**Requested Changes:**

1.	The comparative experiments in this paper are not sufficient.

**Strengths And Weaknesses:**

1.	The motivation is clear and the route is sensible.
2.	The completeness of the paper is good, not only has the experiment but also provides the corresponding theoretical proof.

---

> ### Author Response · Authors · 2023-10-16
> **Response to Reviewer**
>
> Thanks for your comments, and we kindly ask that the reviewer add more detail about their specific concerns so we may further address them.
>
> **The comparative experiments in this paper are not sufficient.**
>
> We include additional experiments with two recent works: FedPAC [1] and FEDORA [2].
>
> FedPAC and FEDORA promote knowledge transfer between personalized models on clients that are similar to each other. However, they can exacerbate the entanglement to spurious features because they encourage the knowledge transfer between clients with similar spurious features while isolating clients from different environments.
>
> | Acc_Worst | MNIST | Coil20 | CelebA | BAR |
> | -------- | ------- | ------- | ------- | ------- |
> | FedPAC | .755 | .859 | .873 | .654 |
> | FEDORA | .721 | .847 | .851 | .643 |
> | Ours  | .870 | .963 | .925 | .730 |
> *Higher is better.
>
> | Acc_Disp | MNIST | Coil20 | CelebA | BAR |
> | -------- | ------- | ------- | ------- | ------- |
> | FedPAC | .236 | .102 | .082 | .122 |
> | FEDORA | .269 | .115 | .109 | .146 |
> | Ours  | .081 | .023 | .002 | .031 |
> *Lower is better.
>
> | Acc_Avg | MNIST | Coil20 | CelebA | BAR |
> | -------- | ------- | ------- | ------- | ------- |
> | FedPAC | .873 | .910 | .914 | .715 |
> | FEDORA | .855 | .904 | .905 | .716 |
> | Ours  | .911 | .974 | .926 | .745 |
> *Higher is better.
>
> **Reference**
>
> [1] Xu, Jian, Xinyi Tong, and Shao-Lun Huang. "Personalized federated learning with feature alignment and classifier collaboration." International Conference on Learning Representations. 2023.
>
> [2] Wu, Jun, et al. "Personalized Federated Learning with Parameter Propagation." Proceedings of the 29th ACM SIGKDD Conference on Knowledge Discovery and Data Mining. 2023.

---

> > ### Comment · Reviewer_rNGy · 2023-10-18
> > **Thanks for the new results**
> >
> > Thanks. A comparison to recent methods helps.

---

### Review · Reviewer_VDA3 · 2023-09-27

**Summary Of Contributions:**

The paper studies the connections between personalization and vulnerability to spurious features in FL. It shows that the personalization strategy such as fine-tuning may worsen the local models by making them more vulnerable to spurious features, which further hurt the generalizability. By exploring the relation between adversarial transferability and the entanglement to spurious features, the paper proposes a method based on adversarial samples that mitigate the effect of spurious features during personalization.

**Audience:**

Yes

**Claims And Evidence:**

No

**Requested Changes:**

1. add more literature reviews and experiments to compare with SOTA methods
2. add more interpretation for the theoretical modeling and the bounds, especially how the bounds are related to the algorithmic design
3. see Weakness section

**Strengths And Weaknesses:**

Strengths:
1. The relation between adversarial transferability and vulnerability to spurious features in FL is interesting and less studied in the literature,
2. The paper makes an effort to provide theoretical insights of the proposed solution.

Weaknesses & questions:

1. Mathematically, the entanglement of a model $w$ to the spurious feature is defined as $w_s$ with $||w_s|| = ||w||cos(\theta)$. I am not sure such quantification is reasonable. More justification is helpful.
2. All theoretical results are limited to the logistic regression model, I wonder whether it can be extended to more general settings such as generalized linear models or non-linear models. If not, what are the challenges?
3. Theorem 2 gives an upper bound of $\theta’$ using arccos function. It is not clear to me if the upper bound is valid. Because the domain of arccos is [-1,1], is the sum of terms inside the parenthesis within [-1,1]?
4. It is not clear to me how the theoretical upper bounds are leveraged for algorithmic design to help generalization. Because the bounds are functions of Lipschitz constant $\rho$, distribution shift, the norm of model parameter, angles between parameters, all are difficult to control. How are these bounds used in the paper?
5. The paper only mentions the related work very briefly in section 2, the comparison is not discussed sufficiently. Moreover, the method is only compared with a few algorithms in the literature, including UW and JTT proposed in 2020, Ditto proposed in 2021. I believe many more personalized methods have been proposed in 2022 and 2023. It is important to compare the proposed method with the state-of-the-art methods.

---

> ### Author Response · Authors · 2023-10-16
> **Response to Reviewer (1/2)**
>
> Thanks for your comments!
>
> **I am not sure such entanglement quantification is reasonable. More justification is helpful.**
>
> Most existing work on disentangling spurious features aims to learn a model with $\mathbf{w}_s = 0$ [1, 2]. Our definition allows a quantitive measurement of the entanglement/disentanglement to spurious features and includes the standard objective as a special case (i.e., $\|w\| = 0$ or $\theta = 0$).
>
> **All theoretical results are limited to the logistic regression model, I wonder whether it can be extended to more general settings such as generalized linear models or non-linear models. If not, what are the challenges?**
>
> The logistic regression model is among the most common choices in prior works on disentangling spurious features [1, 2], and we find that the insight from the model is general – and applicable to more complicated non-linear models. Specifically, we conducted extensive empirical evaluations with various non-linear models showing the effectiveness of the approach. Nevertheless, we agree with the reviewer that extending the theoretical result to generalized linear models or non-linear models is important and interesting for future work.
>
> **Theorem 2 gives an upper bound using arccos function. It is not clear to me if the upper bound is valid. Because the domain of arccos is [-1,1], is the sum of terms inside the parenthesis within [-1,1]?**
>
> The sum of terms inside the parenthesis is within [-1,1] because the codomain of the cosine function in Equation (23), proof of Theorem 2, is [-1, 1].
>
> **It is not clear to me how the theoretical upper bounds are leveraged for algorithmic design to help generalization.**
>
> We briefly discussed the connection between the theoretical analysis and the algorithmic design at the end of Section 4. In the revised version, we have added an additional explanation of our algorithmic design methodology in Section 5:
>
> In Section 4, Theorem 1 suggests that the accuracy disparity is upper bounded by the Lipschitz constant, our entanglement measure, and the distribution shift measured in Wasserstain-1 distance. Here, our entanglement measure is composed of a norm term and an angle term. Then, in Corollary 1, we show that the disparity deviation between a pair of models is bounded by the difference between their entanglement measures. Further, Theorem 2 shows that maintaining the adversarial transferability can help reduce the difference between the angle terms in the entanglement measures – thus illustrating how we can limit the disparity deviation by maintaining adversarial transferability.
>
> However, this angle-based connection may break due to the change of norm terms in the entanglement measure. Indeed, we found that only maintaining adversarial transferability is ineffective in limiting disparity deviation (Section 3). Therefore, our algorithmic design aims to (1) maintain the adversarial transferability and (2) stabilize the norm terms during personalization. The first part of our design updates the personalized (local) model to make consistent (i.e., incorrect) predictions for adversarial examples that are generated with respect to the global model. This approach results in a model that maintains adversarial transferability. The second part of our design aims to stabilize the weight norm of a personalized model (i.e., the weight norm does not significantly increase or decrease, Figure 4). With stabilized weight norms, maintaining adversarial transferability can help reduce the angle between weights of personalized and global models (Theorem 2).
>
> The Lipschitz constant and the distribution shift are not among the objectives of our algorithmic design. They indicate the difficulty of obtaining a small accuracy disparity across environments.

---

> ### Author Response · Authors · 2023-10-16
> **Response to Reviewer (2/2)**
>
> **It is important to compare the proposed method with the state-of-the-art methods.**
>
> We include additional experiments with two recent works, FedPAC [3] and FEDORA [4], which promote knowledge transfer between personalized models on clients that are similar to each other. However, they can exacerbate the entanglement to spurious features because they encourage the knowledge transfer between clients with similar spurious features while isolating clients from different environments.
>
> | Acc_Worst | MNIST | Coil20 | CelebA | BAR |
> | -------- | ------- | ------- | ------- | ------- |
> | FedPAC | .755 | .859 | .873 | .654 |
> | FEDORA | .721 | .847 | .851 | .643 |
> | Ours  | .870 | .963 | .925 | .730 |
> *Higher is better.
>
> | Acc_Disp | MNIST | Coil20 | CelebA | BAR |
> | -------- | ------- | ------- | ------- | ------- |
> | FedPAC | .236 | .102 | .082 | .122 |
> | FEDORA | .269 | .115 | .109 | .146 |
> | Ours  | .081 | .023 | .002 | .031 |
> *Lower is better.
>
> | Acc_Avg | MNIST | Coil20 | CelebA | BAR |
> | -------- | ------- | ------- | ------- | ------- |
> | FedPAC | .873 | .910 | .914 | .715 |
> | FEDORA | .855 | .904 | .905 | .716 |
> | Ours  | .911 | .974 | .926 | .745 |
> *Higher is better.
>
> **The paper only mentions the related work very briefly in section 2.**
>
> We have added more discussion about other personalization techniques:
>
> “FedPAC (Xu et al., 2023) and FEDORA (Wu et al., 2023) further integrate the idea of clustering and fine-tuning: they enable knowledge transfer between personalized models on clients that are similar to each other. There are also approaches investigating fine-tuning using sub-networks (Shamsian et al., 2021) or using a k-nearest-neighbor (kNN) classifier as an augmentation to a local model (Marfoq et al., 2022). Our work focuses on limiting the entanglement of personalized models during fine-tuning via efficient regularization. Therefore, we will compare our approach with other fine-tuning approaches that focus on the loss function design (Li et al., 2021; Xu et al., 2023; Wu et al., 2023). The sub-network and kNN approaches are not direct competitors to our approach, and their combination with our work can be interesting for future work.”
>
> **Reference**
>
> [1] Rosenfeld, Elan et al. “The Risks of Invariant Risk Minimization.” International Conference on Learning Representations. 2021.
>
> [2] Wang, Haoxiang et al. “Provable Domain Generalization via Invariant-Feature Subspace Recovery.” International Conference on Machine Learning. PMLR, 2022.
>
> [3] Xu, Jian, Xinyi Tong, and Shao-Lun Huang. "Personalized federated learning with feature alignment and classifier collaboration." International Conference on Learning Representations. 2023.
>
> [4] Wu, Jun, et al. "Personalized Federated Learning with Parameter Propagation." Proceedings of the 29th ACM SIGKDD Conference on Knowledge Discovery and Data Mining. 2023.

---

> ### Author Response · Authors · 2023-12-01
> **Message to Reviewer**
>
> Dear Reviewer VDA3,
>
> If you have any further concerns, please let us know.

---

### Review · Reviewer_x621 · 2023-10-02

**Summary Of Contributions:**

In this work, the authors propose studying a new problem in federated learning, namely how personalization relates to distributional robustness in federated learning. The authors empirically show how personalization can overfit to locally spurious features and how this relate to adversarial transferability. Based on these observations, the authors propose new personalization method to overcome such difficulty.

**Audience:**

Yes

**Claims And Evidence:**

No

**Requested Changes:**

- Improve the presentation to clearly write out the setup, motivation of setup, and definitions.
- Discuss how this work related to DRO research and DRO research's application to personalized FL.

**Strengths And Weaknesses:**

Strength:
- This paper is tackling a novel problem.
- The proposed method seem to achieve stronger empirical performance compared to prior baselines.

Weakness:
- The paper is quite hard to read. There are a lot of names and concepts in Section 1 and 3. For example, what does it mean to say 'two environments whose combination is the global environment'? What is the formal definition of entanglement and environment? A lot of the definitions are in Section 4, which causes a lot of troubles for readers to understand the paper. Some other examples include not defining $W_1$ until its last appearance in the appendix. From my point of view, the presentation needs a lot of improvement.
- The motivation of problem setup is a bit unclear to me. It seems that the authors want to build some connection between personalization and distribution shift. However, the authors seem to implicitly assume that although the distribution shift for each client individually, the global distribution remains the same. This seems to be an assumption that is too strong (e.g. I don't think that's typically true for temporal distribution shift). Could the authors provide a few examples / justifications on why such scenario is practical?
- Fig 1a seems to be a natural outcome of ERM: the model wouldn't generalize to out-of-distribution samples under ERM objective. Personalization just seems to be a special case of this. As a result, a large body of discussion with prior efforts on distributional robust optimization is missing. I'm also curious about how applying vanilla DRO training while finetuning could affect the behavior.
- Theoretical results are hard to parse. What do we learn from Theorem 1 and 2 and how do they help in designing the method in Section 5. These details seems to be missing.
- The algorithm is extremely costly as it requires to compute the adversarial example at every iteration. This is equivalent to the time of federated adversarial training. The training time could be multiple times more than that of Ditto, which similarly uses the L2 regularization term but not the adversarial example.

---

> ### Author Response · Authors · 2023-10-16
> **Response to Reviewer (1/2)**
>
> Thanks for your comments!
>
> **The paper is quite hard to read. There are a lot of names and concepts in Section 1 and 3. A lot of the definitions are in Section 4, which causes a lot of troubles for readers to understand the paper.**
>
>
> We have improved the presentation: “two environments whose combination is the global environment” -> “the global environment is a mixture of the two environments.”
>
> The formal mathematical definition of entanglement is in Section 4.2, and the formal mathematical definition of environment is in Section 4.1.
>
> We have added the definition of $W_1$, the Wasserstein-1 distance, to its first appearance in Section 4.3 and the table of notations in the Appendix.
>
> Thank you for pointing this out. We apologize for missing these definitions and hope the changes have improved the presentation.
>
> **The motivation of problem setup is a bit unclear to me. Could the authors provide a few examples / justifications on why such scenario is practical?**
>
> The standard setting of federated learning assumes that each client has a different data distribution [1, 2]. We explicitly define a global distribution as the union of the local distribution of each client. We have removed the “temporal distribution shift” to avoid confusion.
>
> In Section 1, we motivate our setting using an example of action recognition in mobile augmented reality applications. For a training set of a specific user, a rocky cliff background can correlate with climbing activity. However, the user may perform climbing activities in other environments (e.g., on an icy cliff) with a deployed model in the future. Then, the (spurious) correlation between the climbing activity and the rocky cliff background may result in an incorrect prediction in the new environment.
>
> **As a result, a large body of discussion with prior efforts on distributional robust optimization is missing. I'm also curious about how applying vanilla DRO training while finetuning could affect the behavior.**
>
> We followed previous works [3, 4] that applied distributional robust optimization (DRO) to disentangle spurious features. The seminal work is the group DRO approach [3], which selectively assigns higher weights to minority groups during training. However, a disadvantage of the group DRO approach is its requirement for precise group annotations [4] (see Table 1 in the submission). A follow-up work, called just-train-twice (JTT), further removed this requirement, which is comparable to our work and is already included in the experiments. Note that our approach does not require group annotations either.
>
> In addition, we found an issue with the approach of “assigning higher weights to minority groups during training” in the federated learning personalization step– when each client has very few samples. The samples from minority groups can be too few for the model to reach a desirable performance even if we assign higher weights to those minority samples (see Section 6.2.2 for more details). In contrast, our approach does not require access to samples from minority groups during personalization – and we find that our approach does not suffer from such a limitation.

---

> ### Author Response · Authors · 2023-10-16
> **Response to Reviewer (2/2)**
>
> **Theoretical results are hard to parse. What do we learn from Theorem 1 and 2 and how do they help in designing the method in Section 5. These details seems to be missing.**
>
> We briefly discuss the connection between the theoretical analysis and the algorithmic design at the end of Section 4. In the revised version, we have added an additional explanation of our algorithmic design methodology in Section 5, as follows:
>
> “In Section 4, Theorem 1 suggests that the accuracy disparity is upper bounded by the Lipschitz constant, our entanglement measure, and the distribution shift measured in Wasserstain-1 distance. Here, our entanglement measure is composed of a norm term and an angle term. Then, in Corollary 1, we show that the disparity deviation between a pair of models is bounded by the difference between their entanglement measures. Further, Theorem 2 shows that maintaining the adversarial transferability can help reduce the difference between the angle terms in the entanglement measures – thus illustrating how we can limit the disparity deviation by maintaining adversarial transferability.”
>
> However, this angle-based connection may break due to the change of norm terms in the entanglement measure. Indeed, we found that only maintaining adversarial transferability is ineffective in limiting disparity deviation (Section 3). Therefore, our algorithmic design aims to (1) maintain the adversarial transferability and (2) stabilize the norm terms during personalization.   The first part of our design updates the personalized (local) model using adversarial examples that are generated with respect to the global model. This approach results in a model that maintains adversarial transferability. The second part of our design aims to stabilize the weight norm of a personalized model (i.e., the weight norm does not significantly increase or decrease, Figure 4). With stabilized weight norms, maintaining adversarial transferability can help reduce the angle between weights of personalized and global models (Theorem 2).
>
> The Lipschitz constant and the distribution shift are not among the objectives of our algorithmic design. They indicate the difficulty of obtaining a small accuracy disparity across environments.
>
> **The algorithm is extremely costly as it requires to compute the adversarial example at every iteration. This is equivalent to the time of federated adversarial training.**
>
> Suppose there are $N$ data samples, and the personalization needs $E$ epochs, our approach only generates $N$ adversarial examples once using the global model (Section 5.1). In contrast, the federated adversarial training generates $N \times E$ adversarial examples. In our experiments, we perform 16 epochs of local personalization steps followed by model selection. In such cases, the overhead of our approach is $\frac{1}{16}$ = 6.25\% of the federated adversarial training.
>
> **Reference**
>
> [1] McMahan, Brendan, et al. "Communication-efficient learning of deep networks from decentralized data." Artificial intelligence and statistics. PMLR, 2017.
>
> [2] Kairouz, Peter, et al. "Advances and open problems in federated learning." Foundations and Trends® in Machine Learning 14.1–2 (2021): 1-210.
>
> [3] Sagawa, Shiori, et al. "Distributionally Robust Neural Networks for Group Shifts: On the Importance of Regularization for Worst-Case Generalization." International Conference on Learning Representations. 2019.
>
> [4] Liu, Evan Z., et al. "Just train twice: Improving group robustness without training group information." International Conference on Machine Learning. PMLR, 2021.

---

> ### Author Response · Authors · 2023-12-01
> **Message to Reviewer**
>
> Dear Reviewer x621,
>
> If you have any further concerns, please let us know.

---

### Comment · Action_Editors · 2023-12-13
**Clarification on Theorem 2**

Sorry to return to the paper after such as long time - I needed to read it carefully myself because the reviews were so borderline.

I would like to ask the authors for a clarification regarding the interpretation of Theorem 2.

If I understand correctly, high $\theta'$ would indicate that the local model is deviating strongly from the global model, possibly due to overfitting to spurious local features. According to the authors' main claim, this would be associated with reduced adversarial transferability, which would manifest as high adversarial transferability loss according to the definition in Eq. (3).

However, according to Theorem 2, high adversarial transferability loss would lead to smaller $\theta'$, in contradiction with the main claim.

I would very much appreciate if the authors could comment on this apparent discrepancy.

---

> ### Author Response · Authors · 2023-12-13
> **Reply to Action Editors**
>
> Thank you for carefully reviewing the submission and providing valuable comments. A typo in Theorem 2 causes the discrepancy between the theoretical result and the main claim. We have fixed the typo and revised the submission accordingly. In the revised version, high adversarial transferability loss would lead to a larger $\theta'$, which is consistent with our claim.
>
> 1. We have corrected a typo in Equation 10, Theorem 2: $\theta' = \mathrm{arccos} \Big(\frac{1}{\epsilon \cdot \bigcirc} \cdot \Big[\epsilon \cdot \square + \diamond {\color{red} + } \mathbb{E}_{\mathcal{D}} [\ell _{g \rightarrow l}  (f_g, f_l, \mathbf{x}, y)]\Big]\Big)$ &#8594; $\theta' = \mathrm{arccos} \Big(\frac{1}{\epsilon \cdot \bigcirc} \cdot \Big[\epsilon \cdot \square + \diamond {\color{red} - } \mathbb{E} _{\mathcal{D}} [\ell _{g \rightarrow l}  (f_g, f_l, \mathbf{x}, y)]\Big]\Big)$. Here, the sign of the expected adversarial transferability loss $\mathbb{E} _{\mathcal{D}} [\ell _{g \rightarrow l}  (f_g, f_l, \mathbf{x}, y)]$ shall be negative.
>
> 2. Such a typo arose when we rearranged the terms in Equation 23, Appendix C. We have added Equation 24 to clarify the rearrangement of terms. We have also improved the readability of the proof of Theorem 2 in Appendix C.
>
> We would be happy to further address any remaining concerns from the editors.

---

### Decision · Action_Editor_MDry · 2023-12-14

**Recommendation:** Accept with minor revision

**Comment:**

In my own reading of the paper, I also found the paper hard to follow. For example, concepts like "adversarial transferability loss" and "adversarial transferability accuracy" are not intuitive, easy to mix and confusing. I found the "adversarial transferability loss" especially confusing, as I would have expected a high value of "adversarial transferability" to indicate high level of transferability, and the word "loss" is simply confusing in this context.

Based on this, I am recommending acceptance with minor revision subject to clarifying the presentation. This would include at least:
1. Make sure that all terminology is clearly explained. If possible, avoid confusing terminology like "adversarial transferability loss/accuracy".
2. Make sure that responses to all relevant questions from the reviewers are reflected in the paper contents. Reviewer x621 in particular raised concern that the relation to DRO, which the reviewer considered crucial for the paper, is not discussed in the text.

When revising the paper, I would like to remind the authors that TMLR **does not have a strict page limit**, so space limitation is not an excuse for omitting some important material or presenting things unclearly.

**Audience:**

The paper makes interesting observations about overfitting to spurious features in federated learning.
All reviewers agree that some members of the TMLR audience would be interested in the findings.

**Claims And Evidence:**

The claims made in the submission are mostly supported by sufficient and clear evidence.
The reviewers raised concerns about poor writing, which makes the paper hard to follow.
I feel the paper would require a revision to improve the clarity of the writing before it can be accepted (see below for details).

---

> ### Author Response · Authors · 2024-01-19
> **Reply to Action Editors**
>
> Thanks for your constructive comments. We have revised and uploaded the paper accordingly.
>
> 1. We changed the “adversarial transferability loss” to “adversarial transferability measure”, which positively correlates with the adversarial transferability. The text, proof, and figures are revised accordingly.
>
> 2. We integrated all the responses into the revised paper. In particular, in the related work section, we added a comprehensive discussion on our approach and the distributional robust optimization (DRO)-based approach.